# Transforming solid-state precipitates via excess vacancies

Laure Bourgeois [1,2✉], Yong Zhang[2], Zezhong Zhang [2,3,4], Yiqiang Chen [2,5] & Nikhil V. Medhekar [2✉]

Many phase transformations associated with solid-state precipitation look structurally simple, yet, inexplicably, take place with great difficulty. A classic case of difficult phase transformations is the nucleation of strengthening precipitates in high-strength lightweight aluminium alloys. Here, using a combination of atomic-scale imaging, simulations and classical nucleation theory calculations, we investigate the nucleation of the strengthening phase θ′ onto a template structure in the aluminium-copper alloy system. We show that this transformation can be promoted in samples exhibiting at least one nanoscale dimension, with extremely high nucleation rates for the strengthening phase as well as for an unexpected phase. This template-directed solid-state nucleation pathway is enabled by the large influx of surface vacancies that results from heating a nanoscale solid. Template-directed nucleation is replicated in a bulk alloy as well as under electron irradiation, implying that this difficult transformation can be facilitated under the general condition of sustained excess vacancy concentrations.

[1] Monash Centre for Electron Microscopy, Monash University, Victoria 3800, Australia. [2] Department of Materials Science and Engineering, Monash University, Victoria 3800, Australia. [3] Present address: Electron Microscopy for Materials Research (EMAT), University of Antwerp, Groenenborgerlaan 171, 2020 Antwerp, Belgium. [4] Present address: Department of Materials, University of Oxford, 16 Parks Road, Oxford OX1 3PH, UK. [5] Present address: Thermofisher Scientific, Achtseweg Noord 5, 5600 KA Eindhoven, The Netherlands. ✉email: laure.bourgeois@monash.edu; nikhil.medhekar@monash.edu

Solid-state precipitation plays a central role in the microstructural development and hence the properties of many materials. Such materials abound, from shape-memory alloys[1], soft nanostructured magnets[2], gemstones[3] and thermoelectrics[4] to high-strength alloys for aerospace applications[5,6]. Amongst the latter, light alloys are particularly dependent on controlled solid-state precipitation: high mechanical strength requires large nucleation rates of precipitates with specific crystal structures, crystallographic orientation and distribution[7]. Unfortunately, the nucleation of these usually metastable precipitates is often difficult and the atomic-scale mechanisms poorly understood, thus hampering efforts towards rational materials design. Particularly puzzling is the case where the difficult nucleation of strengthening phases is preceded by the easy nucleation of another phase with strong structural similarities, typically a coherent precipitate[7]. Why does the first phase not constitute a template for the second, thus facilitating its nucleation? Such a scenario was suggested many decades ago for numerous Al alloy systems[8,9], but only in the last few years has experimental evidence emerged demonstrating the possibility of nucleation directly on coherent precipitates[10,11]. To date, the factors dictating whether a template-directed transformation pathway is favoured or not remain unknown. In particular, it is unclear whether the governing factor is interfacial energy or strain, an important factor in solid-state nucleation. In addition, the role of lattice defects such as dislocations and vacancies is not understood.

In an attempt to answer these fundamental questions, we chose to investigate the θ″-to-θ′ transformation in the binary alloy Al-1.7at.%Cu. This alloy is often regarded as the textbook system for describing precipitation (or age) hardening[12,13]. It is the alloy system used by Guinier and Preston in their seminal studies[14,15] and has been the basis for many commercial aerospace alloys[16], starting with the Wright Brothers' first flight[17]. This alloy owes its precipitation hardening to the decomposition of an aluminium-copper solid solution $\alpha_{Cu}$ into a series of Cu-rich precipitates. Whereas coherent precipitates (GP zones and the θ″ (Al$_3$Cu) phase) nucleate and grow readily, if unaided the main strengthening phase θ′ (Al$_2$Cu) does not[7], regardless of temperature. This is surprising because there exist strong structural similarities between these different phases[7,12,18]. Especially intriguing is the existence of a small region of θ″ structure at the semi-coherent interface of θ′ precipitates, which suggests that the θ″-to-θ′ transformation takes place during growth or nucleation[19]. Yet this transformation is only very rarely observed to involve whole precipitates[11].

Here we report direct and rapid nucleation of the θ′ phase as well as of a precipitate phase which we denote η′, on pre-existing θ″ precipitates. We describe this nucleation pathway as template-directed, as it involves a precursor phase (θ″) that serves as a structural template for the nucleated phases. Whereas nucleation of the θ′ phase is slow and sparse when the bulk alloy is subjected to a conventional heat treatment, we show that it is rapid and abundant when the heat treatment is applied to a sample with one of its dimensions at the nanoscale. We also reveal the critical role of lattice defects, and in particular vacancies, created from nearby surfaces in enabling template-directed nucleation. These findings, therefore, have important implications for precipitation mechanisms in nanoscale or nanostructured materials, as well as in conditions associated with large numbers of lattice defects such as materials far-from-equilibrium or subjected to extreme levels of deformation or intense ion irradiation.

## Results

### Template-directed nucleation in a nanoscale thin sample.
The atomic structures of the θ″ and θ′ phases are shown in Fig. 1a, b,

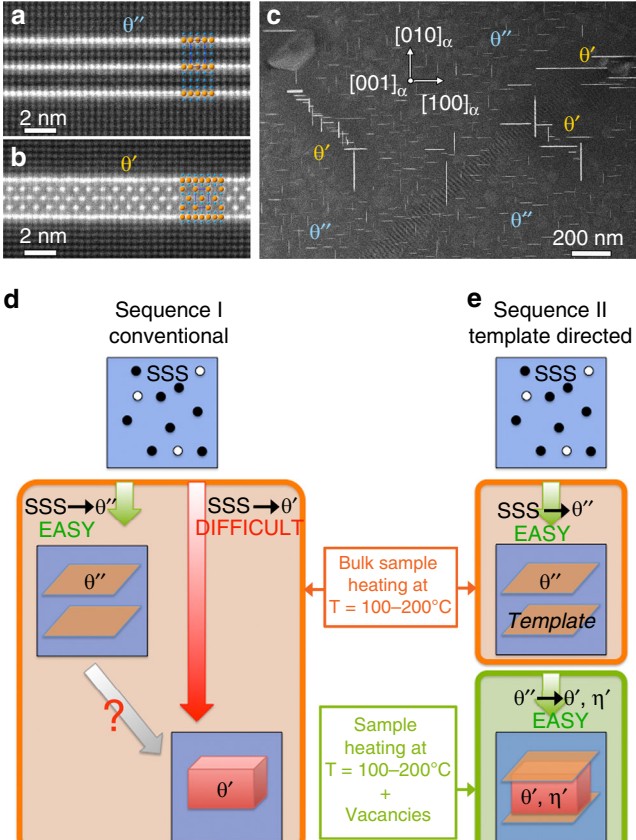

**Fig. 1 Microstructure and crystal structure of precipitate phases involved in conventional and template-directed nucleation. a** Typical θ″ precipitate and **b** θ′ precipitate in a conventionally heat-treated alloy, as viewed along <100>$_\alpha$ by HAADF-STEM imaging. The crystal structures are overlaid, with Cu and Al atoms represented as orange and blue spheres, respectively. **c** Microstructural overview of a Al-1.7at.%Cu alloy conventionally heat-treated for 24 h at 160 °C, showing heterogeneous regions of θ″ and θ′ precipitates. **d** Schematics showing the nucleation sequence starting from a supersaturated solid solution (SSS) in conventional heat treatments and **e** in heat treatments enabling template-directed nucleation (TDN) of the existing phase θ′ and a new phase η′ on the template phase θ″. As will be shown, vacancies are the critical factor required for TDN.

respectively, as imaged by high-angle annular dark field scanning transmission electron microscopy (HAADF-STEM), with the corresponding crystal structures overlaid. The image contrast is dominated by atomic number (Z-)contrast, such that projected regions with a greater density of heavier atomic number element like copper will look bright. The bright dots in Fig. 1a, b therefore, correspond to Cu-rich atomic columns parallel to the viewing direction of <100>$_\alpha$, where α represents the face-centred cubic aluminium matrix (FCC, $a_\alpha = 4.05$ Å). The θ″ phase is coherent with the aluminium matrix α, and consists of Cu atoms that have substituted for Al atoms. The Cu atoms in θ″ arrange as single parallel {002}$_\alpha$ planes separated by three Al {002}$_\alpha$ planes (see Fig. 1a), resulting in a composition of Al$_3$Cu. In contrast the θ′ phase has a structure that is distinct from FCC aluminium. It is tetragonal with lattice parameter $a_{\theta'} \sim a_\alpha = 4.05$ Å[20] and forms as thin platelets along the {200}$_\alpha$ planes, ensuring near-full coherence with the matrix (see Fig. 1b).

A typical heat treatment, or what we will call a conventional heat treatment, consists of applying to a bulk alloy a solution treatment (high-temperature heating) and a quench to produce a solid solution supersaturated in solute and vacancies, followed by

isothermal ageing at a moderate temperature (typically 100–350 °C) to trigger the decomposition of the solid solution into solid-state precipitates[7,12]. The difficult nucleation of θ′ is reflected in a low precipitate number density, as can be seen in Fig. 1c. The bright rod-shaped image features in Fig. 1c correspond to plate-shaped θ″ and θ′ precipitates viewed edge on along two of their three possible <001> orientations. θ′ precipitates form dense colonies[21] in a sea of smaller θ″ precipitates. Under continued ageing, these colonies slowly grow, progressively consuming the θ″-rich regions. In this situation θ″ precipitates appear to dissolve and be replaced by the tetragonal phase θ′, which has a lower free energy[22]. We have found no evidence of a direct structural transformation from θ″ to θ′. The precipitation sequence (Sequence I, Conventional) can, therefore, be summarised as shown in Fig. 1d.

In order to examine nucleation of the θ′ phase at high spatial and temporal resolutions, we performed in situ heating experiments in the transmission electron microscope (TEM). This approach has recently been successful in characterising the evolution of precipitates embedded in a crystalline matrix at near atomic scale[23–25]. Using this method we achieved high nucleation rates of the θ′ phase as well as of a new precipitate phase (called η′), directly on pre-existing θ″ precipitates. This template-directed nucleation mechanism is shown schematically as Sequence II, in Fig. 1e, and is now described.

To achieve template-directed nucleation, or TDN, we first heat-treated a bulk sample via a conventional ageing treatment. The resulting microstructure is shown in Fig. 1c. A thin specimen suitable for TEM examination was then fabricated. Such samples are typically 10–200 nm thick over a region of ~20 μm extending radially away from the central hole. In situ heating in the TEM were then carried out for a series of different temperatures and holding times, from 30 to 200 °C and 5 min to several hours. TDN was observed to take place at in situ heating temperatures as low as 120 °C following ~8 min (see Supplementary Fig. 1). Figure 2 shows TDN observed in a 37 ± 5 nm-thick region having undergone in situ heating at 160 °C for different times. A region rich in coherent θ″ precipitates is shown in Fig. 2a, with three areas of that region displayed at high magnification in Fig. 2d–f. The characteristic Cu multilayered structure of θ″ is evident. In situ heating in the TEM at 160 °C for 10 min resulted in the dramatic transformation revealed in Fig. 2b and enlargements (Fig. 2g–i): nucleation is observed inside all θ″ precipitates shown in Fig. 2a except one, with three examples highlighted in Fig. 2g–i. A previously unreported precipitate phase, denoted η′, is observed in many locations (see Fig. 2g, h). Other θ″ precipitates are found to fully transform to the θ′ phase (see Fig. 2i and Supplementary Fig. 2k, l). Following longer heating times such as a further 60 min at 160 °C (see Fig. 2c), most η′ precipitates were replaced by the θ′ phase—see for instance Fig. 2j, k, while a lone previously untransformed θ″ precipitate has now also transformed to the θ′ phase (Supplementary Fig. 2m, r). The quick and numerous formation of the η′ phase after 10 min heating at 160 °C and its progressive replacement by the θ′ phase is shown quantitatively in Fig. 2m–o. Generally, η′ precipitates nucleated in greater numbers but did not grow as long as θ′ precipitates.

The following key points must be stressed regarding these structural transformations. The transformations occur after only 10 min at 160 °C when heating a thin TEM specimen (i.e. sample with one nanoscale dimension), and are observed at temperatures as low as 120 °C (see Supplementary Fig. 1). In contrast, no such transformations are observed in bulk samples, even after 3 hours at 160 °C (in addition to the original ageing of 24 h at 160 °C—see Supplementary Fig. 3). These transformations are not electron-beam induced and occur on the entire TEM specimen (at least its electron transparent regions). It should also be emphasised that

nucleation takes place within the coherent θ″ precipitates. The θ″ Cu layers are retained within the new structures, except in the case of the three-layered precipitates shown in Fig. 2i and Supplementary Fig. 2r. Moreover, an unexpected phase, η′, forms in abundance. Supplementary Fig. 4 and Supplementary Table 1 demonstrate that this phase is based on the bulk thermodynamically stable phases $\eta_1$ and $\eta_2$, of chemical formula AlCu[26]. The bulk $\eta_1$ phase is stable above ~560 °C, with a crystal structure only solved relatively recently[26]. The η′ precipitate structure is energetically stable at 0 K, with a formation energy per atom slightly less than that of a $1.5c_{\theta'}$-thick θ′ embedded in Al (see Supplementary Table 2). Note that, although the low-temperature form $\eta_2$ has a lower formation energy than $\eta_1$ in the bulk, as expected at 0 K, the situation is reversed when these phases are in thin precipitate forms: the experimentally observed η′, based on $\eta_1$, is energetically preferred over $\eta_2′$, based on $\eta_2$ (see Supplementary Table 2). This can most likely be attributed to a lower interfacial energy and/or strain energy of $\eta_1$ in precipitate form compared with $\eta_2$.

The absence of (1) TDN and (2) the η′ phase from conventionally heat-treated bulk alloys may at first seem surprising when one considers the strong structural relationships between the different precipitate phases, as illustrated in Fig. 2p. Apart from a significant lattice expansion of ~10% in the [001] directions of θ′ and η′ (i.e. the vertical direction of Fig. 2p) compared to $[001]_{\theta''}$, the phases θ′ and η′ differ from the coherent phase θ″ solely in the middle Al/Cu layers (layer A for θ″ and layers ABA for θ′ and η′—see red dashed ellipse). The three phases share the same coherent interfaces with the Al matrix α. In other words, the Cu layers of θ″ can be regarded as templates for the θ′ and η′ phases. These structural similarities suggest that TDN should be associated with a lower interfacial energy and therefore a lower nucleation barrier relative to nucleation directly from the solid solution, and consequently a greater nucleation rate. It seems therefore surprising that TDN should be observed with very high nucleation rates in in situ heating experiments [this work, 9, 11] or in pre-deformed alloys[27], but not in conventional heat treatments. The difference is indeed striking, with in situ heating resulting in very high nucleation rates, typically 1–2 nuclei per θ″ precipitate within minutes at 160 °C. This is in contrast to nucleation rates of nearly zero in conventional heat treatments at the same temperature and longer times (see Supplementary Fig. 3). Additionally, the average precipitate radius at nucleation, i.e. the critical radius for nucleation, is significantly smaller for TDN at 160 °C (~1 nm) compared with nucleation in the bulk using conventional heating conditions (≥3 nm at 100 °C—see Supplementary Fig. 6). The answer lies in a more detailed examination of the TDN rate in in situ heating experiments, as illustrated in Fig. 3a: the number of nuclei observed in projection in the TEM is constant regardless of sample thickness. In addition, the number of nuclei per template precipitate is at least one for thicknesses below ~30 nm, which is similar to the average template precipitate length of ~20 nm (see Fig. 3b). Direct evidence that the nuclei form very close to the surface was obtained from a high-angle tilt series in BF-STEM mode (see Fig. 3c and Supplementary Fig. 7). In Fig. 3c it can be seen that TDN will occur at the cut surface of θ″ precipitates, but not in precipitates that are fully embedded in the aluminium matrix. Note that the cut surface of θ″ precipitates is located at the aluminium-surface oxide interface, with the oxide layer >5 nm thick (see Supplementary Fig. 8), not at the specimen surface-vacuum interface. TDN, therefore, occurs fully in the solid state. These observations imply that the main factor responsible for the high nucleation rate is linked with proximity to the specimen surface. As we shall now explain, we propose this factor to be none other than vacancies.

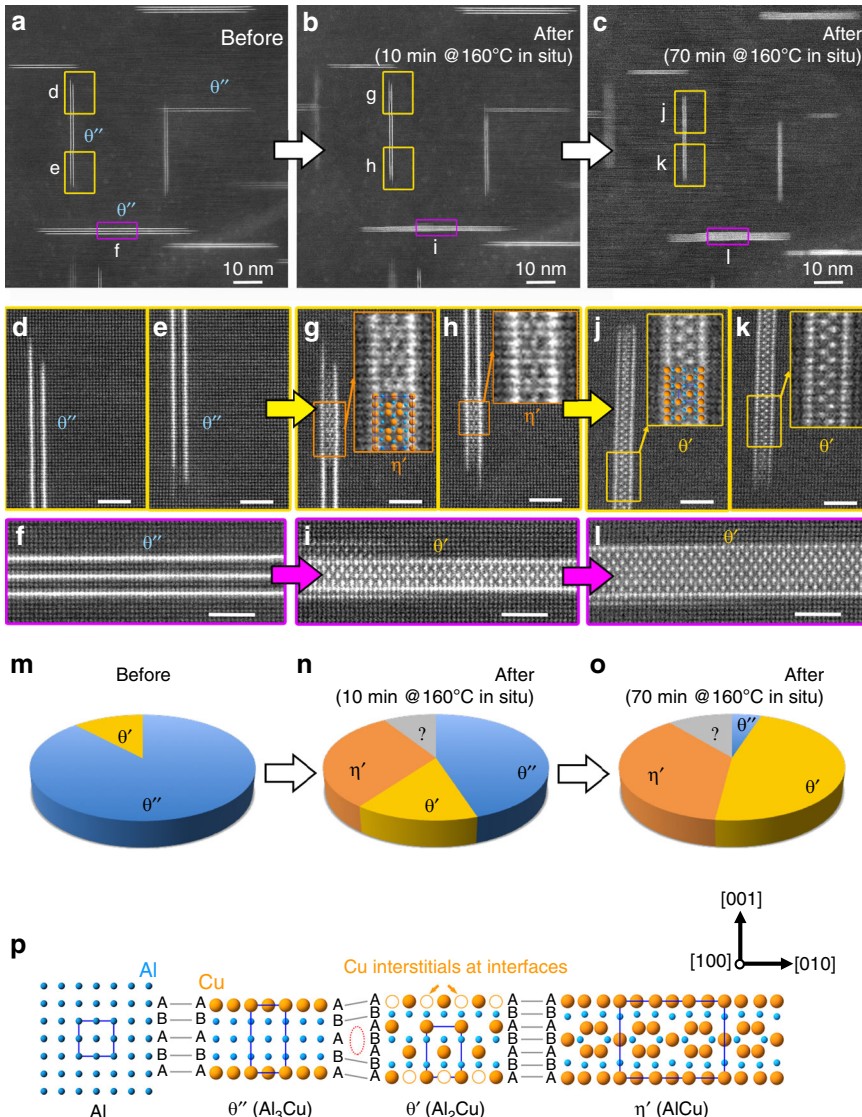

**Fig. 2 Template-directed nucleation (TDN) of the θ′ phase and an unexpected phase, which we denote η′, on the template θ″ phase, as promoted by in situ heating in the TEM. a**, **d**–**f** Show a region and its θ″ precipitates following a conventional bulk heat treatment of 24 h at 160 °C, before in situ heating. **b**, **g**–**i**, **c**, **j**–**l** Show the same region and precipitates after in situ heating for 10 min and 70 min at 160 °C, respectively. The θ′ and η′ phases have nucleated within the θ″ precipitates. The scale bars in **d**–**l** correspond to 2 nm. **m**–**o** Show the proportion of precipitate type (θ″, θ′ and η′) for each of the three conditions and for the same sample region, of thickness 37 ± 5 nm and containing >100 precipitates. The η′ phase nucleates in greater numbers early on, but is progressively replaced by θ′. The grey portion marked "?" refers to nuclei, θ′ or η′, that could not be identified. **p** illustrates the strong structural relationships between the different phases (viewed along [100]), except for atomic planes indicated by a dotted ellipse.

Surface effects can promote solid-state precipitation via a variety of processes: precipitation at the surface[28,29], precipitation at surface-generated dislocations due to heat-induced bending of the thin specimen, precipitation kinetics enhanced by solute flux towards the surface or vacancies flowing to or from the surface[28]. As just mentioned, TDN occurs fully in the solid state and preferentially at the aluminium-aluminium oxide interface. Our experiments (Supplementary Fig. 1) show that heat-induced strain does not play a major role in achieving TDN: TDN still occurs when the sample is heated very slowly, thus undergoing minimal strain caused by the temperature (Supplementary Fig. 1b). Additionally, θ′ and η′ precipitates occur in all their possible orientations (see Supplementary Fig. 4), indicating that surface-induced strain is not a key element in favouring TDN. Furthermore, long-range solute diffusion is not required for TDN; in fact existing θ″ precipitates constitute a reservoir of locally available Cu solute that negates the need for long-range

solute diffusion for phases richer in Cu than θ″. This analysis leaves vacancies as the only factor able to explain the remarkable enhancement of the nucleation rate exhibited by TDN in a nanoscale specimen.

**Atomic scale mechanisms of the θ″-to-θ′ transformation**. We will now focus on the θ″-to-θ′ transformation and show how it can be promoted by vacancies. The atomic scale mechanisms depicted in Fig. 4a, b are proposed for this transformation based on our experimental observations (see also Supplementary Note 2 and Supplementary Figs. 13, 14 for more details). Here a one unit-cell θ″ precipitate provides a template for a 1.5-$c_{θ′}$-thick θ′ nucleus. No additional Cu is required for the transformation. On the other hand, vacancies enable nucleation of the θ′ phase both kinetically and thermodynamically. In a similar way to a sliding puzzle, a single vacancy in the middle Al lattice plane of θ″ (pink

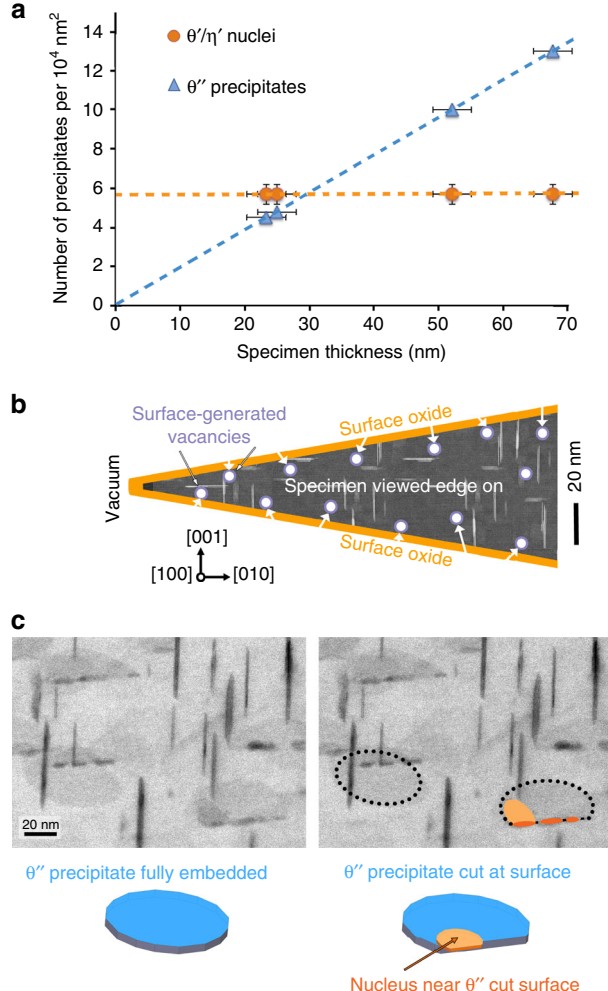

**Fig. 3 TDN is enabled by vacancies generated at the surface of the nanometre-thick specimen. a** Displays a plot of the number of θ′/η′ nuclei and θ″ precipitates per unit area of sample in plan view as a function of sample thickness, with error bars showing the standard deviation. The fact that the number of θ′/η′ nuclei per unit area of sample is independent of thickness indicates that TDN nucleation is a surface or near-surface effect. **b** Shows a schematic illustration of the TEM specimen viewed edge on, with the thinnest section on the left, as imaged by HAADF-STEM. A significant proportion of θ″ precipitates are either intersecting or within 5 nm of the surface and will, therefore, encounter surface-generated vacancies (shown as white disks in greatly exaggerated size). Note that the specimen's wedge angle is depicted here much larger than in reality for the sake of clarity. **c** Shows one frame of a BF-STEM tilt series (see Supplementary Fig. 7), revealing that TDN occurs at the θ″ precipitate cut surface where it meets the specimen surface oxide.

circle in Fig. 4a) should greatly facilitate rearrangement of that plane into the stacking configuration of θ′ (Supplementary Fig. 13c). Our density functional theory (DFT) calculations indicate that a vacancy at such a location will be favoured energetically by $0.08 \pm 0.02$ eV compared with a matrix vacancy at thermal equilibrium. A vacancy in the Al planes immediately below a Cu plane (pink circle in Fig. 4a and Supplementary Fig. 13f) should lower the energy barrier associated with the atomic shift of a Cu atom (see orange arrows, Fig. 4b and Supplementary Fig. 13f). Vacancies can also play a thermodynamic role through lowering the misfit strain of the nucleus, as illustrated by the green circles in Fig. 4b and Supplementary Fig. 13e. This was supported by calculations of the energetics for systems

with and without vacancies, using deep neural network potentials (DNNP) and DFT (Supplementary Fig. 14 and Supplementary Table 3). In particular, we found that vacancies segregating close to the semi-coherent and coherent interfaces will result in a lowering of the total energy of the system, compared with matrix vacancies (Supplementary Fig. 14 and Supplementary Table 3). In other words, the nucleus will act as a vacancy sink. The maximum volume of vacancies at the coherent interfaces able to lower the total energy of the system was comparable to the compressive misfit strain of a 1.5-$c_{\theta'}$-thick θ′ nucleus, further supporting this scenario (see Supplementary Note 3.4).

It should be noted that earlier models[9,30] already included vacancies in the θ″-to-θ′ transformation mechanism (Supplementary Fig. 15). However our experimental observations (Supplementary Fig. 12) and calculations (Supplementary Table 4) favour a mechanism involving as few vacancies as possible, as is the case for the model proposed herein. This mechanism can be readily extended to the nucleation of the η′ phase. In this case, additional Cu atoms will be required; these may be supplied from the θ″ template via short-range diffusion.

**Classical nucleation theory calculations.** In an attempt to quantify why TDN is promoted in a nanoscale specimen but not in bulk samples subjected to conventional heat treatments, we calculated the energy barrier of nucleation for different situations using classical nucleation theory (CNT)[12]—see Supplementary Note 3. Only nucleation of the θ′ phase is considered, as all its energy parameters required are well known[22,31], in contrast to the unexpected η′ phase. According to CNT, nucleation of a phase will only take place for precipitates of that phase large enough to overcome the nucleation energy barrier. This thermodynamic barrier results from the cost associated with forming the new precipitate within the matrix. This energy cost consists of the interfacial energy between matrix and precipitate, and misfit strain[12]. The nucleation rate is determined by the thermodynamic barrier just mentioned as well as by a kinetic term that includes how fast solute is made locally available and how fast the structural transformation associated with nucleation can be carried out. As pointed out earlier, solute (Cu) is in plentiful supply locally in the form of θ″ precipitates, or may not be needed at all (as for nucleating 1.5-$c_{\theta'}$-thick θ′); hence the first term of the kinetic contribution to the nucleation rate will not be limiting in the case of TDN. On the other hand, accomplishing the structural transformation will be greatly facilitated by vacancies, as demonstrated above. Therefore a large vacancy flux will considerably increase the kinetic term of the nucleation rate, which will in part explain the very high nucleation rate observed for TDN nucleation at temperatures as low as 120 °C (see Supplementary Fig. 1). However, the much smaller critical radius of nucleation observed for TDN in situ compared with conventional heat treatments (see Supplementary Fig. 6) implies a lower thermodynamic barrier for TDN in situ. Our DNNP simulations revealed that vacancies segregating around the nucleus will indeed lower the energy of the nucleus. A similar finding is obtained using CNT, as shown in the following.

We calculated the energy barrier of a plate-shaped precipitate of θ′ phase nucleating according to three scenarios. The first scenario deals with direct nucleation from a supersaturated solid solution representing conventional heat treatments in the bulk (see Fig. 4c). The second scenario involves nucleation directly on the template phase θ″, without the thermodynamic effect of vacancies being taken into account (see solid curves in Fig. 4d). The third scenario adds to the second scenario the thermodynamic effect of vacancies, namely vacancies relieving misfit strain (see dashed curves in Fig. 4d).

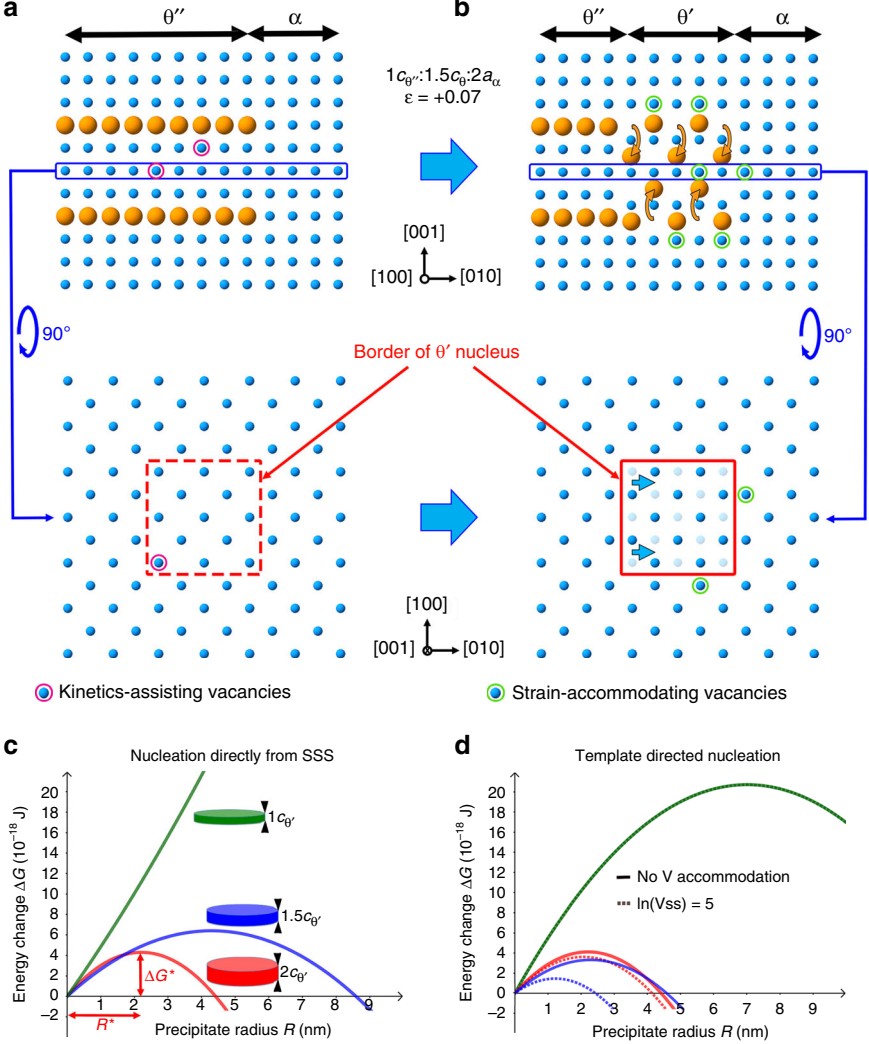

**Fig. 4 TDN of θ′ on θ″ is promoted both kinetically and thermodynamically by vacancies. a, b** Propose atomic mechanisms for how vacancies can assist the θ″-to-θ′ transformation for a 1.5$c_{θ′}$ θ′ precipitate. There is no net change in Cu solute content. Vacancies can accelerate the kinetics by facilitating structural changes (pink circles) and lower the thermodynamic barrier to nucleation by relieving the compressive misfit strain (green circles). The light blue disks in **b** show Al atoms before the transformation. Plots of the computed nucleation energy barrier for the θ′ phase based on Classical Nucleation Theory (CNT) are presented in (**c, d**) for the SSS nucleation and TDN processes, respectively. Green, blue and red curves correspond to θ′ nuclei 1 unit cell (1$c_{θ′}$), 1.5$c_{θ′}$ and 2$c_{θ′}$ thick, respectively. $R^*$ is the critical radius and $ΔG^*$ the nucleation barrier for the red curve. Whereas 2$c_{θ′}$ thick nuclei are favoured over 1.5$c_{θ′}$ thick nuclei in the SSS scenario (**c**), the reverse is found for TDN (**d**). This reversal is strongly enhanced when strain accommodation by vacancies (V) is accounted for and the vacancy supersaturation *Vss* is high, such as for *ln(Vss)* = 5 (**d**).

The model used for these calculations and its sensitivity to various parameters are described in detail in Supplementary Note 3. We considered the three thinnest θ′ plate configurations, i.e. 1$c_{θ′}$, 1.5$c_{θ′}$ and 2$c_{θ′}$—see Supplementary Fig. 16 and insets in Fig. 4c. A θ′ nucleus forming directly from a supersaturated solid solution (SSS) was found to slightly favour a 2$c_{θ′}$ plate thickness both in terms of smaller critical radius $R^*$ and lower nucleation barrier $ΔG^*$. In contrast, a single-unit-cell-thick precipitate is highly unlikely to nucleate on the basis of its extremely large $R^*$ and $ΔG^*$ (Fig. 4c). These results agree qualitatively with calculations from the previous studies[31]. However, the present work finds a larger $R^*$ value of ~2 nm, which is in reasonable agreement with our experiments which show $R^*$~3 nm (see Supplementary Fig. 6b) but is double that calculated in earlier work[31]. Note that the calculated nucleation energy barrier $ΔG^*$ and critical radius for nucleation $R^*$ for θ′ shown were found to change little across temperatures in a range of 100–200 °C so that the calculations plotted in Fig. 4c can be compared with

experiments conducted at different temperatures within that temperature range (see Fig. 2, Supplementary Figs. 1 and 6a). Experiments also revealed the existence of 1.5$c_{θ′}$-thick θ′ nuclei in conventional heat treatments (see Supplementary Fig. 9), in contradiction with all previous reports. This is not consistent with CNT calculations for an SSS transformation, as they show that the nucleation barrier and critical size of 1.5$c_{θ′}$-thick nuclei are significantly greater than for the preferred nucleus configuration of 2$c_{θ′}$ (Fig. 4c).

It is well known experimentally that below ~220 °C and in a conventional heat treatment for bulk material, the θ″ phase will form before θ′ (see refs. [20,21] and Fig. 1). This is because the nucleation barrier for θ″ is much lower than that of θ′, as it only requires Cu diffusion and direct replacement of Al atoms by Cu solute. Therefore the probability that θ′ will nucleate from the SSS before θ″ will be very low (unless assisted by dislocations or specific trace alloying elements[22,32–34], as observed experimentally (Fig. 1 and Supplementary Fig. 3). Once θ″ has formed in

pure Al-Cu, there are two possibilities for θ′ to nucleate without the assistance of dislocations. Firstly, θ′ might nucleate straight from the supersaturated solid solution. However, SSS will have a significantly reduced supersaturation that our calculations suggest to be too low to enable nucleation (see Supplementary Note 3.1.2 and Supplementary Fig. 21). Alternatively θ′ may nucleate directly from θ″, using the templating afforded by the similarity in structures (see Figs. 2p and 4a, b). As mentioned earlier, the second mechanism has never been directly observed in conventional heat treatments. The calculated energy changes associated with templating on θ″ (solid curves in Fig. 4d) imply that, contrary to nucleation directly from a reduced SSS, this mechanism is viable. This is because templating enables a reduction in interfacial energy (see Supplementary Note 3.2). In TDN, $1.5c_{θ′}$-thick nuclei are now as equally likely as $2c_{θ′}$-thick nuclei. It is also worth noting that $1c_{θ′}$-thick nuclei are now possible (though with a low probability of occurrence due to the large barrier $ΔG^*$ and critical radius $R^*$). Such nuclei have been observed in Al-1.7Cu with trace alloying additions of Au[22] but never elsewhere, including in our in situ TEM heating experiments. Our simple CNT-based model of TDN is able to reflect the experimental observation that the probability of forming a $1c_{θ′}$-thick nuclei becomes significant in the presence of Au atoms in the θ′ structure but not in pure Al-Cu (see Supplementary Note 3.3).

To incorporate the thermodynamic role of vacancies into CNT calculations, we adapt the theoretical framework set out by previous authors[8,35]—see Supplementary Note 3.4. Here vacancies are modelled to relieve not only positive volumetric strain but also the strain associated with the change in atomic plane stacking during nucleation (see Fig. 4a, b, Supplementary Figs. 13, 14 and Supplementary Eq. 10). The vacancy contribution to reducing the energy barrier to nucleation consists of two terms (see Supplementary Eqs. 10, 14). One term describes the reduction in misfit strain energy resulting from vacancies segregating around the nucleus (Supplementary Fig. 14). This term includes both a reduction in compressive volumetric misfit strain and a lowering in the semi-coherent interfacial energy through its structural component. A second term will be an additional driving force associated with the removal of excess vacancies from the aluminium matrix to the nucleus. The calculated energy changes as a function of precipitate radius for this situation, i.e. TDN and vacancy-relieved strain, are presented as dashed curves in Fig. 4d. Both $ΔG^*$ and $R^*$ can be seen to decrease significantly in the case of a vacancy supersaturation, $V_{SS}$, for the two thinnest nuclei. Nuclei of $t = 1.5c_{θ′}$ thickness remain the most favoured configuration, because only these display a positive volumetric strain. This is in agreement with our experiments (see Supplementary Fig. 9). Note that the numbers shown for the vacancy supersaturation in Fig. 4d are the natural logarithm of $V_{SS}$ (i.e. $\ln(V_{SS})$—see Supplementary Note 3.4). A reasonably small value for $\ln(V_{SS})$ of ~5 (i.e. $V_{SS} ~ 150$) is required for $R^*$ to drop from 2.3 to ~1.3 nm, i.e. close to the experimentally observed value (Supplementary Fig. 6). A reduction in semi-coherent interfacial energy brought about by vacancy segregation at the semi-coherent interface will further decrease $R^*$ and $ΔG^*$ (Supplementary Fig. 25b).

Whereas a large $V_{SS}$ is present in the very early stages of ageing in a bulk sample thanks to the large concentration of quenched-in vacancies (based on an equilibrium vacancy concentration of $8 × 10^{-5}$ at the solution treatment temperature of 525 °C compared with $3 × 10^{-8}$ at 160 °C), most of these excess vacancies will be lost to sinks such as grain boundaries whilst the θ″ phase precipitates first. By the time the system starts lowering its total energy by nucleating θ′, $V_{SS}$ will be low. This will result in nucleation barriers for θ′ as shown by the solid curves in Fig. 4d,

which, as mentioned above, do not reflect the experimentally observed critical radii of ~1 nm.

A natural question arising is how can proximity to a surface be associated with a large $V_{SS}$. At thermal equilibrium, vacancies will naturally be present in the bulk crystal. The sample surface and internal defects like grain boundaries and dislocations will act as vacancy sources and sinks[7,12]. This is because vacancies have a significantly lower formation energy at these locations compared to within the bulk[36,37]. A solid with one or more nanoscale dimensions, such as our very thin TEM sample, will present two large surfaces very close to the internal space of the alloy, and these surfaces will constitute the main sources and sinks of vacancies. First-principles calculations[37] yield vacancy formation energies as low as 0.15 eV at low-index surfaces of aluminium, compared with 0.65 eV in the bulk. Such low vacancy formation energies will result in vacancy concentrations at or near the surface greater than in the bulk by many orders of magnitude. In our experiments, the aluminium surface is covered by a thin (~5–10 nm) oxide so the exact value of the surface vacancy formation energy or whether vacancies also originate at the Al-oxide interface are unclear. We suggest that some of these surface or near-surface vacancies will travel into the bulk and effectively constitute excess vacancies just below the surface, leading to an effective vacancy supersaturation $V_{SS}$. A simple model where the vacancy supersaturation is estimated by the ratio of the equilibrium vacancy concentrations at the surface and in the bulk, suggests values as high as 10, as measured by $\ln(V_{SS})$. These values are much greater than the $\ln(V_{SS})$ values considered in our nucleation energetics calculations (see Fig. 4d). However, this high $V_{SS}$ should be limited to very close (i.e. within 1–2 nm) of the surface and rapidly decrease when moving into the bulk of the matrix. This provides an explanation for the strong confinement of TDN near the surface of the thin sample (Fig. 3).

**Template-directed nucleation in bulk or irradiated aluminium.** To further test the idea that TDN in pure Al-Cu requires a plentiful supply of vacancies, we performed the following unusual heat treatment to a bulk alloy sample. Standard ageing at 160 °C for a range of times (5–24 h) was carried out in order to grow θ″ precipitates. The alloy was then lightly deformed (3–10%) to introduce dislocations, then heated again at 160 °C for a short time (typically 0.3–8 h) with the aim to generate vacancies from climbing dislocations. As shown in Fig. 5a–e, this three-stage process yielded TDN of θ′ (Fig. 5b) as well as the η′ phase (Fig. 5c–e). We must stress again that the η′ phase is never observed in conventional heat treatments, whereas its frequency of occurrence is over 5% in the three-stage heat treatment. It is likely that in some cases dislocations interacting directly with θ″ precipitates, and not vacancies, enabled the θ″ to θ′ transformation (see Supplementary Fig. 10). Indeed, dislocations have long been recognised to constitute preferential heterogeneous nucleation sites for the θ′ phase[21]. However, two facts strongly suggest that many precipitates nucleated thanks to vacancies. Firstly, very few precipitates were found directly associated with dislocations (Supplementary Fig. 10). Secondly, the η′ phase does not form in conventional heat treatments; in this case there are some dislocations (formed during quenching) but very few excess vacancies following θ″ precipitation.

Further proof of the crucial role of vacancies in TDN is our observation that the electron beam in the transmission electron microscope induces the θ″-to-θ′* transformation, where θ′* is a disordered analogue of θ′ (see Fig. 5f–j). It is well known that high-energy electrons generate vacancies and interstitial defects[38], and clearly these defects enable a difficult nucleation process via a template structure. Additional experiments (see Supplementary

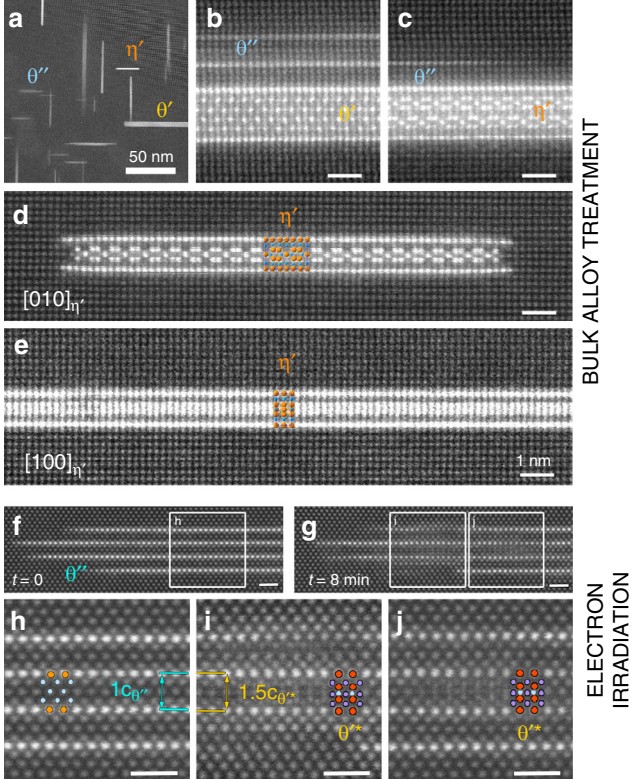

**Fig. 5 TDN can be replicated in a bulk alloy and in samples subjected to a high-energy electron beam.** TDN is observed in a bulk sample subjected to a bulk heat treatment designed to release extra vacancies (**a–e**), and in a thin specimen irradiated by 300 keV electrons (**f–j**). **a** Shows a typical view of the microstructure along <100>$_\alpha$ following ageing at 160 °C, a small deformation and further ageing at 160 °C. Many θ″ precipitates have transformed into the θ′ and η′ phases. **b**, **c** Present evidence that θ′ and η′ nucleated inside a θ″ template precipitate. **d**, **e** Show well developed η′ precipitates along their [010] and [100] directions, respectively, with the crystal structure overlapped. **f** Shows a pristine θ″ precipitate viewed along <110>. **g** Shows the same θ″ precipitate after being subjected to the electron beam in scanning mode for 8 min. **h–j** are enlargements of the boxed regions highlighted in (**f**, **g**). Additional atom columns are clearly visible in (**i**, **j**) that are consistent with the crystal structure of θ′ viewed along <110>, as illustrated by the overlapped crystal structures of θ′ (red = Cu, purple = Al) and the matrix (light blue). However, the nuclei are in fact a disordered analogue of θ′, θ′* (see Supplementary Fig. 11). Unlabelled scale bars correspond to 1 nm.

Fig. 11) clarified the crystal structure of θ′*, demonstrating that it is closely related to beam-damaged θ′ in which some Cu atoms are displaced into vacant interstitial positions. The θ′* structure thus looks identical to that of θ′ when viewed along <110> (Supplementary Fig. 11) but reveals considerable disorder when viewed along <100>. Notwithstanding this structural disorder, θ′* possesses the same ABA-to-AAA stacking change that is characteristic of θ′ and that requires injection of vacancies to facilitate nucleation.

## Discussion

In this work, we reveal how to promote a usually difficult solid-state transformation between precipitate phases in the textbook alloy Al-1.7at.% Cu. This transformation is described as template-directed nucleation (TDN), whereby the template is a coherent precipitate phase that forms easily, here the θ″ phase, and nucleation is that of a phase that usually nucleates with difficulty, here θ′, or an unexpected phase, here η′. TDN may thus be

regarded as a special type of heterogeneous nucleation. In this mechanism, the required structural similarities between template and nucleating phase offer the possibility to nucleate unexpected and potentially useful precipitate phases. We propose an explanation as to why TDN is never seen in conventional heat treatments of bulk alloys: it requires injection of a large amount of excess vacancies after the formation of the coherent phase. TDN is demonstrated to occur in very different conditions: proximity to a free surface (as in a nanoscale sample), releasing of vacancies from dislocations in a strained sample, or intense electron irradiation in a transmission electron microscope. Classical Nucleation Theory calculations and atomistic simulations by density functional theory and deep neural network potentials show that vacancies segregating around the nucleus will lower the thermodynamic barrier to nucleation, as observed experimentally.

There remain several unanswered questions concerning template-directed nucleation. Firstly, why does a phase associated with a very different part of the equilibrium Al-Cu phase diagram, η′, nucleate at high rates? One should emphasise that the extra Cu needed for this phase richer in Cu originates from the pre-existing coherent precipitates. Our DFT calculations also suggest that the η′ phase presents a significantly lower interfacial energy with the matrix and/or strain energy when comparing $1c_{\eta'}$-thick η′ with $1.5c_{\theta'}$-thick θ′ (see Supplementary Table 2 and ref. [39]). This may explain why η′ is observed to nucleate with roughly similar probability compared with θ′ (see Fig. 2), but is then replaced by the expected phase θ′ as the system moves towards equilibrium under continued heating. A detailed thermodynamics study of the η′ phase should clarify this point. Secondly, it is tempting to identify the structure of the complex semi-coherent interface between θ′ and the matrix[19] as the θ″-to-η′-to-θ′ transformations fixed in time, with half a unit cell of η′ corresponding to the $\theta'_t$ region described in ref. [19]. Our experimental observation that η′ is a precipitate phase intermediate between θ″ and θ′ would indeed support this hypothesis. Lastly, why does η′ appear to be promoted by extra vacancies but not by dislocations or a strain field? It may be that dislocations enable the greater barrier to θ′ nucleation to be crossed and thus a greater lowering of the total energy of the system. Answering these questions, as well as ascertaining the precise role of vacancies in those phase transformations, will require an extensive study using sophisticated atomistic calculations of the energy barriers for each atomic step.

Clearly, heating of a nanoscale-thin TEM specimen results in precipitation that does not mirror the conditions of heat-treating a bulk sample. Therefore extreme care must be taken when attempting to extrapolate heating experiments of TEM samples into what might happen in the bulk. Nevertheless, as shown here, such experiments can be useful for discovering alternative nucleation and growth pathways as well as previously undiscovered phases. We applied the knowledge deduced from the in situ heating experiment to obtain similar phase transformation behaviour (i.e. TDN and the η′ phase) in a bulk alloy. Another example of a new phase found in a well-known alloy system using this method is the ζ phase in Al-Ag[10]. This points to the universality of not only using in situ heating to gain knowledge on phase transformations, but also of TDN and the role of vacancies. Other systems where coherent precipitates form first, whether in metallic alloys[7] or even minerals[3] are numerous and may well exhibit TDN and are worth exploring. These mechanisms are also likely to take place in real-life situations such as fatigue, creep, irradiation[40] and novel processing treatments[41] of engineering alloy components, or in the far-from-equilibrium conditions associated with 3D-printing[42]. Finally, our approach may offer a strategy for alloy design based on injection of vacancies, thus avoiding the reliance on costly or environmentally detrimental microalloying additions[43].

## Methods

**Alloy fabrication**. An alloy with nominal composition of Al-1.7 at.%Cu was investigated. It was melted from high purity elements (Al: 99.92%, Cu: 99.8%). The cast ingots were homogenised for 48 h at 520 °C and hot extruded at 450 °C into plates 14 mm thick and 60 mm wide. The extrusion ratio was 16:1. The actual composition of the alloy was determined spectroscopically to be Al-1.63Cu (0.03Si, 0.01Fe) at. %, where Si and Fe are impurity elements. These impurities were concentrated at grain boundaries.

**Heat treatments and TEM sample preparation**. The cold-rolled plates were cut into disks 3 mm in diameter and 0.5 mm in thickness. These were then heat treated according to a conventional age hardening regime: solution treatment in a nitrate salt bath for 30 min at 525 °C, followed by a water quench to 20 °C, then isothermal ageing in an oil bath for 2 h or 24 h at 150 °C, and a final water quench to 20 °C. The heat-treated disks were ground to a thickness of 0.15 mm and twin-jet electropolished in a solution of 33% nitric acid and 67% methanol at −20 °C using a voltage of 13 V.

In situ heating employed a furnace holder (Gatan 652 Tantalum double-tilt holder) in a JEOL 2100 F field-emission gun transmission electron microscope. This holder can reach the desired temperature within 1 min.

**Scanning transmission electron microscopy (STEM)**. High-angle annular dark field (HAADF) and bright-field (BF) scanning transmission electron microscopy (STEM) were performed on a dual-$C_s$-corrected FEI 80-300 Titan[3] operated at 300 kV. A 15 mrad convergence semi-angle was used, corresponding to ~1.2 Å resolution, with a collection inner semi-angle of 55 mrad and an outer collection angle of about 200 mrad for HAADF, and a collection inner semi-angle of 13 mrad for BF. No image processing was performed on the images other than minor contrast and brightness adjustments. Preliminary investigations were made on a JEOL JEM 2100F operated at 200 kV.

**Density functional theory (DFT) calculations**. DFT calculations were performed using the VASP plane-wave pseudopotential code[44]. All calculations were carried out under the generalized gradient approximation (GGA), and used the Projector Augmented Wave potentials supplied with the code[45]. Geometry relaxations were performed with an energy cut-off of 500 eV, allowing ionic positions as well as supercell vectors to relax until the Heynman-Feynman forces were less than 0.01 eV. Å$^{-1}$. The convergence of the relevant energy differences with respect to energy cut-off, k-point sampling and supercell size was better than 2 meV.

The θ″, θ′ and η′ precipitates were modelled using supercells in which the precipitates were surrounded on each (001) side by Al (representing an infinitely wide two-dimensionally coherent precipitate surrounded by Al matrix), and containing the equivalent of 20 to 22 {200} planes. The effect of the large matrix was incorporated by constraining the basal parameters of these supercells to the DFT-optimized lattice parameter of Al (where all lattice parameters and all internal coordinates were relaxed). The defect energy of Cu, $E_{Cu}$, was calculated by replacing a single Al atom by a Cu atom in a $3 \times 3 \times 3$ supercell of pure Al containing 108 atoms. The formation energy of the different phases is given in two forms: (1) $E_f$, relative to the bulk FCC phases of pure Al and pure Cu; (2) $E_f^{Cu}$, relative to the bulk FCC phases of pure Al and pure Cu, and to the defect energy of Cu.

**Deep neural network potential (DNNP) simulations**. To simulate the aggregation of vacancies around embedded θ′ precipitates, large supercells (~500 atoms) are inevitable, which make direct DFT calculations very time-consuming. Here we adopted the deep potential molecular dynamics method that can achieve the accuracy of quantum mechanics in molecular dynamics simulations[46,47]. A potential was built by training a deep neural network model with 69200 DFT calculated configurations (32–185 atoms per configuration, labelled with energies and forces). These configurations include Al-Cu solid solutions, θ′/Al interfacial structures, η′/Al interfacial structures, bulk η′, bulk θ′ and other critical but not well-named structures that the authors have observed by TEM in the past decade. Different numbers and configuration of vacancies were also included in these configurations. After-training validation indicates that the root-mean-square errors of the trained deep neural network potential are 4 meV per atom (energy) and 80 meV Å$^{-1}$ relative to DFT-calculated data sets never seen during training. Such accuracies are sufficient for deep neural network based potentials and should give predictions comparable to typical DFT calculations[48]. The internal coordinates inside the supercells with vacancies located near θ′ precipitates were relaxed using the conjugate gradient method (energy tolerance: $10^{-10}$, force tolerance: $10^{-8}$) in the large-scale atomic/molecular massively parallel simulator (LAMMPS) software package (http://lammps.sandia.gov)[49]. As in the DFT simulations, the supercell lattice parameters were kept constant. The validity of our Deep Neural Network Potentials was confirmed through DFT-calculated vacancy formation energies of vacancies close to a θ′ precipitate, whereby both methods yielded the same trends in energies.

**Classical nucleation theory (CNT) calculations**. The thermodynamic barrier to nucleation of θ′ precipitates was calculated using Classical Nucleation Theory[12], assuming the nucleus has a disk shape. Full details of the calculations can be found in the Supplementary Note 3.

## Data availability

The datasets generated during and/or analysed during the current study are available from the corresponding author on reasonable request.

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

## Acknowledgements

The authors are indebted to Matthew Weyland for his expert advice on aberration-corrected scanning transmission electron microscopy. L.B. would like to acknowledge initial discussions with B.C. Muddle and J.F. Nie many years ago regarding the possible thermodynamic role of vacancies in solid-state precipitation. The authors acknowledge funding from the Australian Research Council (LE0454166, LE110100223), the Victorian State Government and Monash University for instrumentation, and use of the facilities within the Monash Centre for Electron Microscopy. The authors thank Flame Burgmann, Dougal McCulloch and Edwin Mayes for access to and assistance at the Microscopy and Microanalysis Facility at RMIT University. L.B. and N.M. acknowledge the financial support of the Australian Research Council (DP150100558). Authors also gratefully acknowledge the computational support from MonARCH, MASSIVE and the National Computing Infrastructure and Pawsey Supercomputing Centre. ZZ and YZ are thankful to Monash University for a Monash Graduate Scholarship, a Monash International Postgraduate Research Scholarship. Z.Z. is grateful for a Monash Centre for Electron Microscopy Postgraduate Scholarship. The authors are grateful to Anita Hill for advice.

## Author contributions

L.B. designed the experiments, carried out most of the microscopy, performed the CNT calculations, and wrote the manuscript. Y.Z. fabricated the samples and performed the Deep Neural Network Potential calculations. Y.Z. and Z.Z. performed the DFT calculations and assisted with crystallographic and thermodynamic analysis. Y.C. carried out some of the microscopy. L.B. and N.M. initiated and supervised the project. All the authors read the paper and discussed the results.

## Competing Interests

The authors declare no competing interests.
