## [Peer Review File · Nature Communications]

Reviewers' comments:

Reviewer #1 (Remarks to the Author):

This paper describes nicely performed microscopy experiments and classical nucleation theory calculations to describe a phenomenon termed "template-directed" nucleation (TDN) in the Al-Cu system. The microscopy images are stunning, and the authors have attempted to present a straightforward treatment of a complex problem.

Big picture, there are major problems with this manuscript. These are detailed in the comments below:

- Excess vacancies facilitate transformations and this is well known and the reason behind quenching of specimens after solution treatment. The title of the paper is somewhat misleading. The word 'Difficult' in the title is not very informative nor descriptive of what they have done. Maybe, the presence of a nanoscale dimension needs to be in the title. Template directed nucleation is already well known. The GP zones form on dislocation loops (discs of vacancies), for example, so the title of the paper comes across as repackaging of known phenomenon.
- At some level, it appears that this manuscript is a long and somewhat rambling description of an experiment gone wrong. The authors started out trying to do in-situ heating experiments to study precipitation in the Al-Cu system. Instead they found that heat-treating a TEM foil does not produce the same precipitation reaction produced in bulk samples. This 'unexpected' precipitate has been seen in other in-situ heating experiments on Al-Cu alloys, but these researchers did a more complete evaluation and description of this phase they have called η' (because of its similarity to a known phase in Al-Cu alloys). The most interesting part of their study was when they were able to form η' in bulk material by lightly deforming to introduce dislocations. This would have been a more attractive study if it centered on those experiments on bulk materials.
- The authors state in the abstract that, "this template-directed....". This is ultimately a speculative statement. This is literally a hypothesis that is proved by eliminating couple of other presented options. The present reviewer believes that indeed the higher concentration of vacancies is a possibility, so is higher diffusion rates of copper without the presence of a higher vacancy concentration. Because the arguments in the paper rest on this speculative hypothesis, we can not stand behind this being a major new discovery in physical metallurgy. All the derivations in the paper can be redone for a higher diffusion coefficient of copper without vacancy arguments. Definitive proof of a higher vacancy concentration (like Positron Annihilation spectroscopy or even indirect measurement like resistivity change) in the nanoscale dimension will be needed before this claim can be considered a breakthrough. Authors observe that the transformation of θ'' to η' starts to occur only at surfaces of θ'' precipitates that are at cut surfaces of their TEM foil. From this observation they propose/conclude that more readily available vacancies are responsible for this unusual transformation. Vacancies are one explanation, but the surfaces also have a different strain state and are a fast diffusion path.
- It is not obvious to us why classical homogeneous nucleation theory is used to explain the formation of η' . What they see is that this new phase does not form independently. It comes about with the rearrangement of solute in an existing phase. Also, it is surprising that the Al_3Cu goes to Al_2Cu with an intermediate phase that is AlCu in composition. Any comments? This does point to kinetics and easy movement of copper atoms by higher vacancy concentrations or just the fact that copper can diffuse faster at the surface of a TEM foil. Again without proof of the higher vacancy concentrations, the arguments seem speculative.

Minor comments to improve the manuscript:

- Page 2 – line 7 – "unfortunately, the nucleation...." One can argue that the overall mechanism of nucleation of the metastable phases in the aluminum-copper system is quite well understood. Indeed, this system constitutes the text book example of the same. The authors allude to the same in the next paragraph.
- Page 2 – line 20. Is there a GP alloy? Unfair to coin a new alloy name if it does not exist. GP zones are a standard term not GP alloy.
- References 16 and 17 need to be interchanged according to the manuscript text. Ref 17 is for

commercial aerospace alloys and 16 for the Wright Brothers crankcase

- Page 2 – line 24. At higher temperature theta prime nucleate and grow readily.
- References 19-23 are not cited in the main manuscript!
- How do we know that the features being referred to in fig 3 are surface oxides?
- Authors argue on Page 8-line 8 that TDN occurs in the solid state preferentially in the Al-alumina interface. Even if this is true, why can the solute flux not be enhanced at this interface as well? In any case, the surface oxide is likely very thin (a few nm according to the authors) so the authors are stretching the definition of bulk in this case.
- Page 11 – line 34 – "...the exact value of the surface vacancy formation..." This statement needs to be clarified, it is not clear what the authors are referring to.
- Page 12 – line 6 – "... aim to generate vacancies from climbing dislocations..." while intersecting dislocations can create jogs and leave vacancy trails, annealing by itself is unlikely to cause dislocation climb (you will need an external stress specially at a temperature of 160 C). This can again be related to dislocations increasing the effective diffusion coefficient of copper. You need to fatigue a sample to generate this excess concentration of vacancies as noted in the recent paper (Reference 44 in your manuscript).
- In the concluding remarks, the authors state that TDN can be regarded as a special type of heterogeneous nucleation. In the opinion of the reviewer, TDN is heterogeneous nucleation.
- Why did the authors perform electron microscopy at 300 kV electrons? Their microscope is aberration-corrected and capable of lower voltages that would produce much less damage in Al.

Reviewer #2 (Remarks to the Author):

The authors provide a wealth of very nice experimental evidence for the role of excess vacancies in the sequence of precipitation in the classical metastable solid-state precipitation system Al-Cu. It is interesting to note that we are still learning about Al-Cu, perhaps the best-documented system of its kind.

The work introduces concept of "template-directed nucleation" and provides strong experimental and theoretical support for the idea.

The authors are careful to point out the limitations of in-situ electron microscopical study of these reactions. They then go on to exploit these limitations to advance their research.

The manuscript is well-crafted and clear; the experimental work is exceptional, and very well-presented.

Reviewer #3 (Remarks to the Author):

The authors have studied Guinier Preston zones in Al-Cu alloys. Using various transmission electron microscopy techniques, they evidenced that heterogeneous precipitation of the stable theta' phase on the metastable theta' phase is enhanced in a thin foil because because of the proximity of the surface which allows. This enhanced nucleation of the stable phase is assumed to derive from the high vacancy supersaturation which exists close to the surface, and the assumption is validated by showing that the same precipitation mechanism exists in other experimental conditions leading also to a vacancy supersaturation (vacancies created by plastic deformation or by electron irradiation). A model based on the classical nucleation theory is then developed to rationalize these experimental observations. I think this is a nice and original work on an important topic, precipitation in an industrial alloys where several competitive phases can appear. From my point of view, the experimental observations are very convincing, but I think that the description of the vacancy contribution and its modeling have to be improved.

Right now, it is really hard, when reading the manuscript and the supplemental materials, to fully understand how vacancies modify the precipitation kinetics. The authors describe two different effects of vacancy absorption by the precipitates: some rearrangement at the interface, maybe leading to partial loss of coherency, and the accommodation of the misfit strain, but these two effects are poorly described. It is actually not clear if these are two different contributions. The authors simply wrote "Here vacancies are modelled to relieve not only positive volumetric strain but also the strain associated with change in atomic plane stacking during nucleation" in the main manuscript, without giving much more explanation. Only Fig. 4d tries to illustrate the physical mechanism behind vacancy absorption. I think this illustration should be improved, maybe by also adding projections in other directions. The authors should think about showing the interface before and after absorption of the vacancies (Fig. 4c). As one effect of the vacancies seems to allow the joining between atomic planes of types A and B, they should find a graphical way to evidence the difference between these planes on all their atomic representations. Finally, they should calculate the vacancy contribution to the precipitate eigenstrain: right now, they only precise these eigenstrains (actually only one component corresponding corresponding to traction/compression perpendicular to the platelet) for perfectly coherent interfaces and it is not clear how the vacancies will modify these eigenstrains.

The modeling of the vacancy contribution to the nucleation driving force is only described at the end of the supplementary materials (section 2.4). This description is too succinct for the reader to judge on its validity. All the vacancy contribution is included in the equation given at the top of page 22, but the authors do not describe how they obtain this equation. Besides, this equation looks very surprising as it is adding a volume and a lateral contribution, with the strain appearing only in the first one and with no parameter involving elasticity. As all the vacancy contribution is included in these equations, the authors are urged to describe how they obtain such an equation and the different physical ingredients it contains. Besides, as this vacancy contribution corresponds to the main result of the manuscript, I do not think that the authors can simply describe it in the supplementary materials: the reader should be able to understand how vacancies impact precipitation and how it can be modeled by only reading the main manuscript.

Other minor comments:

- the authors wrote p. 6 "Secondly, these transformations are not electron-beam induced, and occur on the entire TEM specimen" This is in apparent contradiction with what is described latter where it appears that the transformation only appear in a thin layer under the thin foil surface.
- when showing that template nucleation occurs close to the surface, the authors directly concluded that this is caused by the vacancies. One could have thought that elastic relaxations caused by the surface could have a role also.

Revised Manuscript

“Enabling Difficult Solid-Solid Phase Transformations through Excess Vacancies” by L. Bourgeois, Y. Zhang, Z. Zhang, Y. Chen and N.V. Medhekar

Reviewers' comments are shown in black and the authors' response to each comment in **blue**.

We would like to thank the three reviewers for their detailed and constructive comments regarding our manuscript. Before separately addressing each point made by the referees, we would like to highlight the main novelty of our work.

By examining the textbook alloy system Al-Cu, we describe how to directly transform an easily formed coherent precipitate into a strengthening precipitate, whose nucleation is usually difficult. We show that the key factor in promoting the phase transformation is the presence of excess vacancies. Excess vacancies are rare once coherent precipitates have formed, but we show different strategies to inject them, thereby causing the desired transformation from coherent precipitates to strengthening precipitates.

Reviewer #1:

This paper describes nicely performed microscopy experiments and classical nucleation theory calculations to describe a phenomenon termed “template-directed” nucleation (TDN) in the Al-Cu system. The microscopy images are stunning, and the authors have attempted to present a straightforward treatment of a complex problem.

Big picture, there are major problems with this manuscript. These are detailed in the comments below:

Excess vacancies facilitate transformations and this is well known and the reason behind quenching of specimens after solution treatment.

Response:

It is indeed well known that excess vacancies assist precipitation by promoting solute diffusion and forming secondary defects as nucleation sites. However, these vacancies will be consumed very quickly during ageing through diffusion to vacancy sinks like dislocations and grain boundaries. This results in the known conventional microstructure of Al-Cu alloys that consists in very sparse semi-coherent θ' precipitates amidst a dense distribution of coherent θ'' precipitates.

The novelty of our work is the demonstration that *vacancies can enable direct phase transformations from coherent precipitates*. The inherently different transformation pathways can also lead to unexpected intermediate phases, like the η' phase in this work. The main role of vacancies is in lowering the transformation strain energy associated with nucleation, in addition to the usually considered kinetic role of vacancies as agents of diffusion. Although related through the presence of excess vacancies, the subject of our work differs significantly from precipitation from a supersaturated solid solution. In our study, the use of pre-existing coherent precipitates as the starting point for

the transformation means that long-range solute diffusion is not a limiting factor. As a result, the transformation is essentially a structural one, with only minor changes in chemistry required. We have highlighted these points further throughout the revised manuscript.

The title of the paper is somewhat misleading. The word 'Difficult' in the title is not very informative nor descriptive of what they have done. Maybe, the presence of a nanoscale dimension needs to be in the title.

Response:

Although TDN was initially found by heating a nanoscale thin sample, it was also promoted in the bulk through specific processing conditions. Therefore we do not wish to use "nanoscale" in the title. However we agree with the reviewer that the title could better convey the nature of our work. The title has been modified to "Transforming solid-state precipitates via excess vacancies".

Template directed nucleation is already well known. The GP zones form on dislocation loops (discs of vacancies), for example, so the title of the paper comes across as repackaging of known phenomenon.

Response:

It is indeed true that precipitation is well known to occur on dislocation loops resulting from the collapse of a vacancy loop in quenching and ageing, e.g. θ' itself (Guyot & Wintenberger, J. Mater. Sci. 9 (1974) 614), γ' (AlAg₂) in Al-Ag as well as γ' and θ' in Al-Cu-Ag -- see our own work Rosalie et al. Phil Mag. 89 (2009) 1267 and Acta Mater. 59 (2011) 7168. However the subject of our work is different from precipitation on dislocation loops formed in the early stages of ageing. Here we examine **precipitation on pre-existing coherent precipitates**, a solid-solid phase transformation pathway rarely observed directly and whose atomic scale mechanisms remain poorly understood.

In addition, we do not claim template directed nucleation to be a new phenomenon, either in the title or elsewhere in the manuscript. What we claim as new here is the need for excess vacancies to accomplish the structural transformation, rather than just accelerate the kinetics, in order to enable template directed nucleation on pre-existing coherent precipitates. Coherent precipitates constitute an easily and quickly formed reservoir of solute atoms for the subsequent nucleation of strengthening precipitates. Our paper focuses on the last step, namely how to get nucleation of strengthening precipitates on pre-existing coherent precipitates.

At some level, it appears that this manuscript is a long and somewhat rambling description of an experiment gone wrong. The authors started out trying to do in-situ heating experiments to study precipitation in the Al-Cu system. Instead they found that heat-treating a TEM foil does not produce the same precipitation reaction produced in bulk samples.

Response:

Based on the structure of our manuscript, we can understand the reviewer's assumption of the history of our work. However we would like to point out that our in situ experiments were initially carried out for a reason different from that assumed: they were an attempt to order disordered θ' -like precipitates nucleated under the electron beam (Fig. 5(f)-(j)).

This 'unexpected' precipitate has been seen in other in-situ heating experiments on Al-Cu alloys, but these researchers did a more complete evaluation and description of this phase they have called eta' (because of its similarity to a known phase in Al-Cu alloys). The most interesting part of their study was when they were able to form eta' in bulk material by lightly deforming to introduce dislocations. This would have been a more attractive study if it centered on those experiments on bulk materials.

Response:

An important message of our work is the use of in situ TEM heating to obtain new information about phases and phase transformation mechanisms, which can then be applied to the bulk alloy. Also we believe that enhanced precipitation in nanoscale samples or near surfaces, as shown in our present work, merits attention, as this is relevant to a whole class of new materials.

The authors state in the abstract that, "this template-directed.....". This is ultimately a speculative statement. This is literally a hypothesis that is proved by eliminating couple of other presented options. The present reviewer believes that indeed the higher concentration of vacancies is a possibility, so is higher diffusion rates of copper without the presence of a higher vacancy concentration. Because the arguments in the paper rest on this speculative hypothesis, we can not stand behind this being a major new discovery in physical metallurgy. All the derivations in the paper can be redone for a higher diffusion coefficient of copper without vacancy arguments.

Response:

First, we would like to address the criticism that our observations can equally be explained by increased Cu diffusion, such as along dislocations or close to the surface. Our main findings are two fold:

(1) A key part of the θ'' to $1.5c_0 \theta'$ or $1c_0 \eta'$ transformation involves **aluminium atoms only** (the stacking change of the middle (002) layer – see Fig. 4, Lines 30-31 Page 8). This structural component can be accomplished via the right dislocation, strain or vacancies. However **faster diffusion of copper will not help**, except hypothetically through the release of vacancies assisting Cu diffusion. But this is exactly the role of vacancies highlighted in our manuscript.

(2) **No change in Cu composition is necessary for the θ'' to $1.5c_0 \theta'$ transformation.** Only atomic scale steps are required for Cu atoms to move in order to accomplish the Cu atom part of the transformation. Therefore the fact that this transformation is usually difficult cannot be explained by slow Cu

diffusion. Conversely, enhanced Cu diffusion cannot explain why this transformation is facilitated since there is no need for enhanced Cu diffusion.

We have highlighted these facts in the revised manuscript via new sub-figures (Fig. 4(a)-(b); also see below) and associated text.

The presence of interstitial Cu at the coherent surfaces of θ' will require some local Cu diffusion, but only within a few atomic distances. As already noted in the original manuscript (Line 11-12, Page 8), the additional Cu can be provided by the template structure θ'' . This can be observed through the shortening of the source θ'' precipitate (see Fig. 2 and Fig. S2), a point that is now stressed in the revised manuscript. This is equally valid for the η' phase.

Whilst the θ'' to $1.5c_0 \theta'$ transformation can be promoted without enhanced Cu diffusion, as just discussed, we agree with the reviewer that there is also enhanced Cu diffusion at the surface. This is a well-known phenomenon mentioned in the original manuscript (Lines 4-7, Page 8, Refs [32-33]) and is evident in the thinnest regions of the TEM foil such as near the edge of a hole where the sample is nanoscale in two dimensions and has access to more vacancies, as shown in a new supplementary figure Fig. S8. As can be seen in this case, θ' precipitates at the aluminium-oxide interface near the hole.

Authors observe that the transformation of θ'' to η' starts to occur only at surfaces of θ'' precipitates that are at cut surfaces of their TEM foil. From this observation they propose/conclude that more readily available vacancies are responsible for this unusual transformation. Vacancies are one explanation, but the surfaces also have a different strain state and are a fast diffusion path.

Response:

As explained above, faster Cu diffusion is not required to explain the θ'' to θ'/η' transformations. More importantly, ***Cu diffusion is not sufficient to accomplish the θ'' to θ'/η' transformation***, which must also include the stacking change of an Al plane – see new Fig. 4(a)-(b).

The reviewer suggests that the different strain state of the sample surface may trigger these transformations. We indeed considered this possibility, as stated on Lines 4-14 of Page 8 of the original manuscript. As noted there, surface strains will be exacerbated by rapid heating of a thin foil. However we found little difference in TDN rates in a thin foil whether heating was rapid or slow – see Fig. S1, implying that surface strains are not the main factor in promoting TDN.

Surface strains will also exist whether the sample is heated or not. Such strains should be directional, and in the case of a flat surface, should be normal to that surface. Therefore if surface strains were to influence near-surface precipitation, one would expect certain variants to be favoured. This is not observed, as shown in Fig. S4 for η' , but is also the case for θ' : all possible variants are observed to nucleate. We therefore conclude that surface strains are not a key factor in promoting TDN in a thin sample. This point is now highlighted in the text.

We would also like to point out that TDN is not only observed near the surface of a thin foil, but also in a slightly deformed bulk alloy and in a TEM specimen subjected to electron irradiation (Figs. 5, S9 and S10 of original manuscript). Regarding electron irradiation in particular, no special strain states or long range Cu diffusion are present and can hence be invoked. The electron beam only has a very local effect, as the transformation was observed when the incident beam was scanned over a region typically 10 nm by 10 nm (this information has now been added to Figs. 5 and S10 in the revised manuscript). As explained on Lines 17-26, Page 12 of the original manuscript, electron irradiation of a metallic sample in a TEM will displace atoms and generate both vacancies and interstitials. It is therefore reasonable to conclude that the transformation is enabled by these defects, and in particular vacancies.

Definitive proof of a higher vacancy concentration (like Positron Annihilation spectroscopy or even indirect measurement like resistivity change) in the nanoscale dimension will be needed before this claim can be considered a breakthrough.

Response:

Like the reviewer we would welcome a more direct proof of the role of vacancies in accommodating the strain associated with nucleation. However we believe this is an endeavour for the future, and cannot be achieved with current experimental techniques. In particular, the technique mentioned by the reviewer, positron annihilation lifetime spectroscopy (PALS) is a very indirect method requiring assumptions and detailed modelling of the size and nature of positron traps. In the system examined in our work, such traps are numerous, whether dislocations, precipitate interfaces, vacancies or vacancy clusters and it would be very difficult to disentangle the contributions from the different types of free volumes that can act as positron traps. This is the reason why the overwhelming majority of PALS experiments on alloys focus on clustering of solute atoms and vacancies, prior to precipitation (eg J. Banhart et al. PRB 83 (2011) 14101). In addition, the spatial resolution of a scanning positron microscope is in the μm range, which is inadequate for tackling our current system – see for example Egger et al. Mat. Sci. Eng. A 387 (2004) 317. Therefore the technique does not allow correlation between detected vacancy clusters and precipitates.

Similarly, resistivity measurements only provide very indirect information, which, given the complexity of our system (vacancies, misfit dislocations at precipitate interfaces), cannot lead to any stronger proof of the presence of vacancies.

In our view, atomic resolution transmission electron microscopy will in future be able to image vacancies and vacancy clusters. Promising observations were reported recently by Sun et al (Science 363 (2019) 972, Ref. 44 in original manuscript) suggesting vacancy-solute clusters. However such images cannot yet be taken as unambiguous evidence of vacancies, as demonstrated by experts in TEM image interpretation (Kim et al. PRX 6 (2016) 41063).

What can be accomplished with current TEM techniques are observations of small dislocations, which may be the result of collapsed vacancy clusters. We used different scanning TEM modes sensitive to such defects in an attempt to determine whether these are present close to nuclei. As described in a new Sec. 2 in the Supplementary Information on atomic models of nucleation, the emission of small vacancy loops should be expected for previously proposed nucleation mechanisms (see new figure Fig. S15). As shown in the new figure Fig. S12, we did not detect any dislocation loops, which eliminates some of the possible nucleation mechanisms.

It is not obvious to us why classical homogeneous nucleation theory is used to explain the formation of θ' . What they see is that this new phase does not form independently. It comes about with the rearrangement of solute in an existing phase.

Response:

As stated on Line 20, Page 8, "only nucleation of the θ' phase was considered, as all its energy parameters required are well known". Furthermore, we only used homogeneous nucleation theory as a comparison to template directed nucleation, which is modelled as heterogeneous nucleation on pre-existing θ'' precipitates (Secs. 2.2-2.4 in original manuscript).

Also, it is surprising that the Al_3Cu goes to Al_2Cu with an intermediate phase that is AlCu in composition. Any comments? This does point to kinetics and easy movement of copper atoms by higher vacancy concentrations or just the fact that copper can diffuse faster at the surface of a TEM foil. Again without proof of the higher vacancy concentrations, the arguments seem speculative.

Response:

We agree with the reviewer that this observation is surprising. On Lines 16-23 Page 14 (original manuscript) we suggest a thermodynamic explanation based on DFT calculations, namely a lower interfacial energy between matrix and η' phase. The source of the extra Cu is the pre-existing θ'' precipitate. Given the small critical radius of the nucleus size (~ 1 nm), only short-range diffusion is required, which will be accelerated by the presence of vacancies. We now stress this point in the revised manuscript.

Minor comments to improve the manuscript:

Page 2 – line 7 – “unfortunately, the nucleation...” One can argue that the overall mechanism of nucleation of the metastable phases in the aluminum-copper system is quite well understood. Indeed, this system constitutes the text book example of the same. The authors allude to the same in the next paragraph.

Response:

The following is known about θ' nucleation according to textbooks: its position in the precipitation sequence, its crystal structure and its need for dislocations or microalloying additions to nucleate at higher rates. However the mechanism of nucleation, ie how the atoms move, is not known. Previous authors have proposed different atomic mechanisms for the θ'' to θ' transformation, which are now summarised pictorially in Fig. S15 and compared with our own model (Figs. 4 and S13). Additional experiments (Fig. S12) and simulations (Fig. S14) lend strong support to our model.

Page 2 – line 20. Is there a GP alloy? Unfair to coin a new alloy name if it does not exist. GP zones are a standard term not GP alloy.

Response:

We replaced the text with "alloy used by Guinier and Preston in their seminal work [14-15]"

References 16 and 17 need to be interchanged according to the manuscript text. Ref 17 is for commercial aerospace alloys and 16 for the Wright Brothers crankcase

Response:

Apologies for the error, which is now fixed.

Page 2 – line 24. At higher temperature theta prime nucleate and grow readily.

Response:

No, nucleation rates of θ' precipitates are still low, even at temperatures higher than 160°C, as expected for a lower driving force for nucleation. This is visible in the figure shown below comparing Fig. 1(c) (24h at 160°C, inset) and the microstructure for 30 min ageing at 350°C. The inset and main image are both at the same scale, and it can be seen that the number density is much greater at 160°C. Also compare Fig. 1(c) with images in the literature, e.g. Fig. 2 in Laird & Aaronson, Trans. Met. Soc. AIME, 242 (1968) 1393 (3h @ 250°C). Note that the figure below is not included in the manuscript as it may detract the reader from the main points of our work.

As mentioned above, the focus of the present work is how to transform the densely distributed θ'' phase into the strengthening phase θ' . Note that since the solvus for θ'' is ~220°C, ageing at higher temperatures will only result in θ' -- hence why we did not consider temperatures above 200°C.

References 19-23 are not cited in the main manuscript!

Response:

Apologies. This mistake has been fixed.

How do we know that the features being referred to in fig 3 are surface oxides?

Response:

We include a new figure in the Supplementary Information showing that the oxide layer is at least 5 nm thick – see Fig. S8.

Authors argue on Page 8-line 8 that TDN occurs in the solid state preferentially in the Al-alumina interface. Even if this is true, why can the solute flux not be enhanced at this interface as well?

Response:

We agree that the Cu solute flux may be enhanced at the Al-oxide interface, as this interface is likely disordered. However one key point of our observations and analysis is that *no change in Cu composition* is necessary for the θ'' to θ' transformation. Only atomic scale steps are required for Cu atoms to move in order to effect the transformation. We have highlighted this fact in the revised manuscript via new sub-figures (Fig. 4(a)-(b) and new associated text.

The point raised by the referee is nicely illustrated by the rapid nucleation and growth of θ' near the edge of a hole drilled in the sample, where the proximity to three surfaces rather than two should lead to a greater vacancy influx and enhanced Cu solute diffusion – see new supplementary figure Fig. S8.

In any case, the surface oxide is likely very thin (a few nm according to the authors) so the authors are stretching the definition of bulk in this case.

Response:

We agree that nucleation at the interface of aluminium and a 5-10 nm thick oxide is not a bulk phenomenon and we never claimed it to be. In fact throughout the manuscript we stressed the near surface and nanoscale nature of the process.

Page 11 – line 34 – “...the exact value of the surface vacancy formation...”
This statement needs to be clarified, it is not clear what the authors are referring to.

Response:

This has been reworded as "the exact value of the vacancy formation energy at the surface or at the aluminium-oxide interface".

Page 12 – line 6 – “... aim to generate vacancies from climbing dislocations...” while intersecting dislocations can create jogs and leave vacancy trails, annealing by itself is unlikely to cause dislocation climb (you will need an external stress specially at a temperature of 160 C). This can again be related to dislocations increasing the effective diffusion coefficient of copper. You need to fatigue a sample to generate this excess concentration of vacancies as noted in the recent paper (Reference 44 in your manuscript).

Response:

We agree with the reviewer that at 160°C dislocation climb will not be a major phenomenon. Some of the θ'' to θ'/η' transformations will have occurred through interaction with a dislocation rather than a small number of vacancies, as shown in Fig. S10 and mentioned on Lines 9-15 Page 12.

As pointed out above, the θ'' to θ' phase transformation is not hindered by the lack of Cu (because there is plenty of Cu in θ'' precipitates as solute reservoirs) but by the stacking change of Al planes, and it is this structural transformation that can be accommodated by dislocations, as is well known (Ref. 15) and was pointed out in the original manuscript. However many of the transformed precipitates were not associated with dislocations (see Fig. S10). Whereas enhanced Cu diffusion will no doubt be a factor in promoting the transformation, it cannot be the sole factor due to the necessity of changing the Al (002) plane stacking.

In the concluding remarks, the authors state that TDN can be regarded as a special type of heterogeneous nucleation. In the opinion of the reviewer, TDN is heterogeneous nucleation.

Response:

We agree with the reviewer that TDN is heterogeneous nucleation, as stated in our original manuscript. However we believe that "heterogeneous nucleation" is a general expression that provides little detail about the mechanism, hence our coining the more descriptive term of template directed nucleation.

Why did the authors perform electron microscopy at 300 kV electrons? Their microscope is aberration-corrected and capable of lower voltages that would produce much less damage in Al.

Response:

It is indeed true that a lower accelerating voltage will result in a lower damage rate in Al. However damage at 300 kV was not a problem in the sense that the pristine structures could be captured easily at atomic resolution; repeated acquisitions at high-resolution were required to result in noticeable beam damage, as evident in Fig.5(g) and Fig. S10. Beam damage was in fact a useful phenomenon showing the transformation from θ'' to a disordered θ' structure at room temperature, i.e. with very little Cu solute diffusion beyond the nanoscale but with the creation of vacancies (see Fig. 5(f)-(j)).

Reviewer #2:

The authors provide a wealth of very nice experimental evidence for the role of excess vacancies in the sequence of precipitation in the classical metastable solid-state precipitation system Al-Cu. It is interesting to note that we are still learning about Al-Cu, perhaps the best-documented system of its kind.

The work introduces concept of "template-directed nucleation" and provides strong experimental and theoretical support for the idea.

The authors are careful to point out the limitations of in-situ electron microscopical study of these reactions. They then go on to exploit these limitations to advance their research.

The manuscript is well-crafted and clear; the experimental work is exceptional, and very well-presented.

Response:

We thank the referee for this positive appraisal of our work.

Reviewer #3:

The authors have studied Guinier Preston zones in Al-Cu alloys. Using various transmission electron microscopy techniques, they evidenced that heterogeneous precipitation of the stable θ' phase on the metastable θ'' phase is enhanced in a thin foil because of the proximity of the surface which allows. This enhanced nucleation of the stable phase is assumed to derive from the high vacancy supersaturation which exists close to the surface, and the assumption is validated by showing that the same precipitation mechanism exists in other experimental conditions leading also to a vacancy supersaturation (vacancies created by plastic deformation or by electron irradiation). A model based on the classical nucleation theory is then developed to rationalize these experimental observations. I think this is a nice and original work on an important topic, precipitation in an industrial alloys where several competitive phases can appear. From my point of view, the experimental observations are very convincing, but I think that the description of the vacancy contribution and its modeling have to be improved.

Response:

We thank the referee for their positive assessment of our experimental work. We also understand their criticism regarding the modelling component of this paper, and thank their constructive suggestions for improvement. Here is a summary of the changes made to address the criticisms of Referee #3.

- (1) We now propose detailed atomic scale mechanisms for the θ'' to θ' transformation. These are shown in a revised Fig. 4 as well as in the Supplementary Information.
- (2) These mechanisms were inferred following additional experiments that failed to show vacancy loops (shown in Supplementary Information).
- (3) In these mechanisms vacancies enable/promote TDN kinetically as well as thermodynamically.
- (4) Additional simulations of the energetics using semi-empirical deep neural network potentials (DNNP) and density functional theory (DFT) are presented, showing that vacancies located near both the coherent and semi-coherent interfaces lower the total energy of the system through strain relief.
- (5) These simulations are shown to compare well with classical elastic theory calculations.
- (6) Our original Classical Nucleation Theory calculations have been improved through the incorporation of more specific strain energy terms.

Right now, it is really hard, when reading the manuscript and the supplemental materials, to fully understand how vacancies modify the precipitation kinetics. The authors describe two different effects of vacancy absorption by the precipitates: some rearrangement at the interface, maybe leading to partial loss of coherency, and the accommodation of the misfit strain, but these two effects are poorly described. It is actually not clear if these are two different contributions. The authors simply wrote "Here vacancies are modelled to relieve not only positive volumetric strain but also

the strain associated with change in atomic plane stacking during nucleation" in the main manuscript, without giving much more explanation.

Response:

The modelling of how vacancies promote TDN has been significantly developed in the revised manuscript, with separate sections and figures for the kinetic and thermodynamic roles of vacancies. Regarding kinetics, we propose that at least one vacancy present in the middle layer of the θ'' phase should reduce the energy barriers for some of the atomic shifts required in the transformation to θ' – see new supplementary figure Fig. S13(c),(f). Regarding thermodynamics (or rather energetics as we neglected entropy effects) we now provide DNNP and DFT calculations showing energy reductions when vacancies segregate near both coherent interfaces (to reduce volumetric strain energy) and near one semi-coherent interface (to reduce semi-coherent interfacial energy) – see new Fig. S14 and Table S3.

Only Fig. 4d tries to illustrate the physical mechanism behind vacancy absorption. I think this illustration should be improved, maybe by also adding projections in other directions. The authors should think about showing the interface before and after absorption of the vacancies (Fig. 4c). As one effect of the vacancies seems to allow the joining between atomic planes of types A and B, they should find a graphical way to evidence the difference between these planes on all their atomic representations.

Response:

New figures (see new Fig. 4(a)-(b) and Supplementary Figs. S13, S14) and new Table S3 now illustrate how vacancies will (1) lower the energy of the system by segregating near both coherent interfaces and one semi-coherent interface (Fig. 4(b), Fig. S13(b),(e), Fig. S14, Table S3), and (2) lower the kinetic barriers for atomic shifts in the θ'' to θ' transformation (new Fig. 4(a)-(b) and Fig. S13(c),(f)). In case (1) an embedded precipitate with and without vacancies near the interfaces is shown as a schematics (Fig. 4, S13) and as DNNP simulations (Fig. S14).

Finally, they should calculate the vacancy contribution to the precipitate eigenstrain: right now, they only precise these eigenstrains (actually only one component corresponding corresponding to traction/compression perpendicular to the platelet) for perfectly coherent interfaces and it is not clear how the vacancies will modify these eigenstrains.

Response:

The tensile/compressive strain along the long dimensions of θ' precipitates was neglected as it has been shown experimentally to be extremely small (e.g. R. Sankaran and C. Laird, *Phil. Mag.* 29 (1974) 179-215); furthermore its sign remains unclear: earlier x-ray diffraction work suggested it was negative, whereas recent density functional theory calculations (e.g. H. Liu, B. Bellón, J. LLorca, *Acta Mater.* 132 (2017) 611-626) and our own electron microscopy observations (unpublished) show it to be positive, albeit very small ($\sim 0.7\%$).

The off-diagonal eigenstrain terms were also neglected, as is commonly the case (see S.Y. Hu, PhD thesis 2004, and also presumably S.Y. Hu et al. *Acta Mater.* 54 (2006) 4699-4707). This can be interpreted as neglecting any shape change other than dilatational. Some authors (H. Liu, B. Bellón, J. LLorca, *Acta Mater.* 132 (2017) 611-626) included significant off-diagonal terms ϵ_{23} and ϵ_{32} of -17% in an attempt to capture the stacking change of the middle Al layer from Al matrix to θ' structure as a shear strain. This approach invoked a lattice correspondence between matrix and product (the θ' phase), in essence assuming the transformation mechanisms. In addition, and as remarked in our original Supplementary Information (p21), the shear strain associated with the stacking change of Al planes will be in part incorporated into the interfacial energy.

We now compare the total residual misfit volumetric strain of an embedded θ' precipitate with and without vacancies segregating near the coherent interface based on continuum mechanics. The volume associated with this strain for a nucleus 1 nm in radius is shown to be close to the total volume of vacancies segregating near the coherent interface of the nucleus found in our atomistic simulations (see Supplementary Information Sec. 3.4).

The case of strain relief at the semi-coherent interface is more complex, owing to the poor match between matrix and precipitate. Modelling the strain arising at this interface is not straightforward, especially given that the interfacial structure has not been fully elucidated. However this strain is likely to be highly localised around the semi-coherent interface thanks to the high coherence along the broad surface of a θ' precipitate. Our new semi-empirical DNNP simulations reveal that vacancies located close to one semi-coherent interface reduce the total energy of the system. In our Classical Nucleation Theory calculations we approximate this as a reduction in semi-coherent interfacial energy – see Sec. 3.4 in the Supplementary Information.

The above information is now included in the Supplementary Information.

The modeling of the vacancy contribution to the nucleation driving force is only described at the end of the supplementary material (section 2.4). This description is too succinct for the reader to judge on its validity. All the vacancy contribution is included in the equation given at the top of page 22, but the authors do not describe how they obtain this equation. Besides, this equation looks very surprising as it is adding a volume and a lateral contribution, with the strain appearing only in the first one and with no parameter involving elasticity. As all the vacancy contribution is included in these equations, the authors are urged to describe how they obtain such an equation and the different physical ingredients it contains.

Response:

We thank the reviewer for the insightful comment. The vacancy terms referred to are chemical free energy terms:

$$\Delta G_{chem}^V = -k_B T \cdot \ln(V_{SS}) \left[\frac{V\varepsilon}{a_0} + \frac{2\pi R}{a_\alpha} \right]. \quad \text{Eq. R1}$$

They only include a strain contribution for the volumetric strain case as a way to calculate the number of vacancies required to accommodate this strain. The second term is somewhat approximate because, as mentioned above, it is difficult to model the strain associated with the semi-coherent interface. Our semi-empirical potential simulations (now shown in Fig. S14 and Table S3) show that our early approximation of vacancies segregating around the perimeter of the precipitate to accommodate the strain associated with the semi-coherent interface was reasonable. These calculations also suggested that the number of vacancies per unit length of semi-coherent interface was probably overestimated. The second term of Eq. R1 has therefore been modified to $\pi R/2a_\alpha$ (see Sec. 3, Supplementary Information).

The reviewer's comment allowed an error in our CNT calculations to be uncovered: the original manuscript failed to take into account the reduction in strain energy resulting from vacancy segregation. This omission is now corrected (see Sec. 3 in the Supplementary Information and all corresponding CNT plots), resulting in even lower nucleation barriers for TDN and vacancy segregation than previously calculated for a given vacancy supersaturation.

Besides, as this vacancy contribution corresponds to the main result of the manuscript, I do not think that the authors can simply describe it in the supplementary material: the reader should be able to understand how vacancies impact precipitation and how it can be modeled by only reading the main manuscript.

Response:

We now include a brief description in the main text of the vacancy energy terms used in our CNT calculations. However the equations are left for the Supplementary Information, along the rest of the detailed description of the CNT modelling.

Petrick et al. have recently proposed* a precise scenario to explain the formation of θ' precipitates from GP zones in presence of a vacancy supersaturation. This scenario looks fully compatible with the experimental observations reported in the submitted manuscript. As pointed in my report, right now the authors clearly evidence the necessity of a vacancy supersaturation for precipitation of the θ' phase but fail to propose a convincing mechanism describing how vacancies promote precipitation. I think the scenario proposed by Petrick et al. could be an important part of the story and the authors should consider this work.

* Scripta Materialia 65, 123 (2019); <https://doi.org/10.1016/j.scriptamat.2019.02.024>

Response:

We are grateful to the reviewer for pointing out this recent article by Petrick et al., which is indeed very relevant to the present work. Petrick et al. propose an ingenious scenario for the GPI zone/ θ'' to θ' transformation based on the segregation of vacancies within the Cu-rich planes of the coherent GPI zones or θ'' precipitates. The geometry is such that no shearing of an (002) Al plane

is necessary for the transformation. However, and this is a point not mentioned by Petrik et al., their mechanism will require emission of those excess vacancies, perhaps as two (002) vacancy loops. A similar mechanism involving vacancy segregation inside a θ'' precipitate and the subsequent emission of a vacancy loop was proposed by Dahmen and Westmacott (Ref. 9 in our original manuscript). We now describe these possible mechanisms as well as our own in a new Section 2 in the Supplementary Information, entitled "Atomic scale mechanisms for the θ'' to θ' transformation". To test which mechanism operates, additional TEM experiments were conducted seeking vacancy loops associated with template-directed nucleation on θ'' precipitates – see Fig. S12. Images taken in bright-field STEM mode should be very sensitive to the presence of such dislocation loops, yet none were observed. This is perhaps not surprising given that these models (now illustrated in Fig. S15(a)-(b)) will introduce additional strain energy from dislocation loops compared with our model (Fig. 4 and S15(c)).

Other minor comments:

- the authors wrote p. 6 "Secondly, these transformations are not electron-beam induced, and occur on the entire TEM specimen" This is in apparent contradiction with what is described latter where it appears that the transformation only appear in a thin layer under the thin foil surface.

Response:

In scanning TEM, only a small region of the sample is subjected to the electron beam. At the lowest magnification used, $10\ \mu\text{m} \times 10\ \mu\text{m}$ were irradiated with an extremely low electron dose, whilst at the magnifications used most commonly typically $200\ \text{nm} \times 200\ \text{nm}$ were irradiated. This left most of the electron thin regions of the sample untouched by the electron beam. Yet TDN was observed everywhere on the sample (at least its electron transparent regions), implying that the electron beam is not responsible for the transformation. On the other hand, the nanoscale nature of the sample must be a key factor in promoting the transformation.

- when showing that template nucleation occurs close to the surface, the authors directly concluded that this is caused by the vacancies. One could have thought that elastic relaxations caused by the surface could have a role also.

Response:

We agree with the reviewer on this point, which was also mentioned by Reviewer #1. As indicated above:

We indeed considered this possibility, as stated on Lines 4-14 of Page 8 of the original manuscript. As noted there, surface strains will be exacerbated by rapid heating of a thin foil. However we found little difference in TDN rates in a thin foil whether heating was rapid or slow – see Fig. S1, implying that surface strains are not the main factor in promoting TDN.

Surface strains will also exist whether the sample is heated or not. Such strains should be directional, and in the case of a flat surface, should be expected to be normal to that surface. Therefore if surface strains were to influence near-surface precipitation, one would expect certain variants to be favoured. This is not observed, as shown in Fig. S4 for η' , but is also the case for θ' : all possible variants are observed to nucleate. We therefore conclude that surface strains are not a key factor in promoting TDN in a thin sample. This point is now highlighted in the text.

Additional changes made by the authors:

In addition to the modifications described above, we corrected a number of typographical and reference errors. Very minor text changes were also made in order to improve descriptions – see manuscript with red highlights.

Reviewers' comments:

Reviewer #1 (Remarks to the Author):

Significant issues were raised in the first review. The authors did a good job of addressing the issues in our first review.

Amit Shyam and Matthew Chisholm

Reviewer #3 (Remarks to the Author):

The authors have addressed all the points raised in my previous report. I nevertheless still have two criticisms on the proposed scenario describing the formation of θ' precipitates from θ' and on the model.

1) The authors criticised the mechanisms previously proposed by Dahmen and by Petrik as such mechanisms should reject vacancies and the authors assumed that these vacancies will cluster to lead the formation of vacancy dislocation loops which are not seen experimentally. On the other hand, when developing their own scenario, they proposed that some vacancies will segregate all around the precipitates to accommodate the precipitate strain (vacancies appearing in green in Fig. 4d). I do not understand why the vacancies needed in the scenario proposed by Dahmen or by Petrik could not play the same role and will necessarily cluster.

Another point that bothers me is the Cu interstitial atoms observed at the θ' interface (Fig. 2p). These extra Cu atoms are never really considered in the model, except when choosing a value for the coherent interface energy. Clearly, Cu is not diffusing interstitially in aluminum, but through a vacancy exchange mechanism. Each time a Cu is deposited in an interstitial site at the θ' precipitate, a vacancy also remains at the interface. The mechanism proposed in the present work should also lead to the creation of extra vacancies in the surrounding of the precipitates. I therefore feel that the mechanisms previously proposed by Dahmen and by Petrik and the one described in the present work are not treated on an equal footing. After reading the manuscript, I should admit that I am not more convinced by one proposition, maybe because classical nucleation theory is not the best tool to address such a question which would be better handled by atomic simulations (calculations of the energy barriers involved in the various steps of the different proposed mechanisms for instance).

2) The authors demonstrated that vacancy supersaturation promotes heterogeneous nucleation by incorporating vacancy contribution to the nucleation energy through the corresponding thermodynamic driving force (reduction of vacancy supersaturation), an elastic contribution (reduction of the precipitate eigenstrain), and also an interface contribution. There are still some parameters of the model I do not fully understand: could the authors be more precise when defining the corresponding three contributions in Eq. 3 of SI? I cannot see what is exactly the elastic energy variation ΔG_{el} and I do not understand where the interface term comes from as the authors wrote just after Eq. 3 "that the semi-coherent interfacial energy remains the same". But, more importantly, this contribution of vacancy supersaturation will also be present for homogeneous nucleation and will also enhance homogeneous nucleation. I don't understand why the authors did not consider this supersaturation also for homogeneous nucleation. At the end, I guess that vacancy supersaturation will generally promote nucleation, either heterogeneous and homogeneous, and that heterogeneous nucleation is favoured over homogeneous nucleation mainly because of elastic and interface effects. This point should be made clearer.

To summarize, the proposed mechanism for the formation of θ' precipitate looks reasonable but is a possibility among others. The developed model allows to incorporate the different ingredients corresponding to this mechanism to lead to a reasonable thermodynamic description,

but one cannot claim that this model demonstrates the validity of this new precipitation mechanism. From my point of view, the question is still open and will require further simulations, probably directly relying on an atomic description.

Revised Manuscript

“Transforming Solid-State Precipitates Via Excess Vacancies” by L. Bourgeois, Y. Zhang, Z. Zhang, Y. Chen and N.V. Medhekar

Reviewers' comments are shown in black and the authors' response to each comment in **blue**.

Reviewer #1:

Significant issues were raised in the first review. The authors did a good job of addressing the issues in our first review.

Response:

We are pleased that the reviewers' issues with the original manuscript have been addressed to their satisfaction.

Reviewer #3:

The authors have addressed all the points raised in my previous report. I nevertheless still have two criticisms on the proposed scenario describing the formation of θ' precipitates from θ' and on the model.

Response:

We thank the referee for reading our manuscript in detail for a second time and asking deep questions about the atomic scale mechanisms.

1) The authors criticised the mechanisms previously proposed by Dahmen and by Petrik as such mechanisms should reject vacancies and the authors assumed that these vacancies will cluster to lead the formation of vacancy dislocation loops which are not seen experimentally. On the other hand, when developing their own scenario, they proposed that some vacancies will segregate all around the precipitates to accommodate the precipitate strain (vacancies appearing in green in Fig. 4d). I do not understand why the vacancies needed in the scenario proposed by Dahmen or by Petrik could not play the same role and will necessarily cluster.

Response:

This possibility that excess vacancies in Dahmen and Petrik et al.'s models (now referred to as “past” or “former” models) may be used to accommodate the strain of the newly formed nucleus instead of dissipating into the matrix is an interesting suggestion. However, an estimate of the number of vacancies involved shows that past models require many more vacancies to be ejected (~150 vacancies) than can be accommodated by a typical θ' nucleus 1 nm in radius (~10 vacancies). Therefore former models will still require ejection of a considerable number of vacancies into the matrix. This arises directly from the far greater number of vacancies required by these former models to accomplish the θ'' to θ' transformation (~150 vacancies) compared to our model (~10 vacancies, for a 1 nm nucleus).

This point is made in an additional paragraph and a new table (Table S4) in Sec. 2 of the Supplementary Information describing estimates of vacancy numbers associated with different scenarios.

Another point that bothers me is the Cu interstitial atoms observed at the θ' interface (Fig. 2p). These extra Cu atoms are never really considered in the model, except when choosing a value for the coherent interface energy. Clearly, Cu is not diffusing interstitially in aluminum, but through a vacancy exchange mechanism. Each time a Cu is deposited in an interstitial site at the θ' precipitate, a vacancy also remains at the interface. The mechanism proposed in the present work should also lead to the creation of extra vacancies in the surrounding of the precipitates.

Response:

This is another interesting point which we did not raise in former versions of our manuscript but is now addressed thanks to the reviewer.

Once θ' has nucleated, via whichever mechanism, it is true that additional Cu atoms migrating to interstitial sites at the coherent interfaces will leave vacancies. Some of them may reduce any compressive strain of the nucleus, but as in the situation described above, their number only need to be ~ 10 for a 1 nm nucleus. Assuming each interstitial Cu atom is associated with one vacancy, ~ 70 vacancies will be available from filling all interstitial Cu positions at the coherent interfaces of a 1 nm nucleus. In our mechanism, these vacancies and solute atoms are likely to come from the surrounding template θ'' precipitate. The resulting structure (leftmost diagram in Fig. S15(b)) is known to be stable according to Petrik et al.'s DFT calculations. This is in contrast with intermediate states rich in vacancies from Dahmen & Westmacott and Petrik et al.'s models (middle diagrams in Fig. S15(a) and (b)) where vacancies must be ejected to reduce the considerable tensile strain. Furthermore, vacancies within the Cu planes of θ'' precipitates cannot be treated as free as matrix vacancies, because an Al atom is unlikely to exchange with one such vacancy: to date no evidence has been found of mixed Cu-Al (002) layers. Nevertheless, one can conceive a situation whereby a vacancy-rich θ'' precipitate adjacent to the transformed area of θ' phase can supply additional vacancies for θ' lengthening, until the entire θ'' precipitate is consumed. However, the study of θ' growth (rather than nucleation) is beyond the scope of the present work.

I therefore feel that the mechanisms previously proposed by Dahmen and by Petrik and the one described in the present work are not treated on an equal footing. After reading the manuscript, I should admit that I am not more convinced by one proposition, maybe because classical nucleation theory is not the best tool to address such a question which would be better handled by atomic simulations (calculations of the energy barriers involved in the various steps of the different proposed mechanisms for instance).

Response:

We agree with the reviewer that more theoretical and computational evidence is required for the correct mechanism to be determined unambiguously. In

particular, we cannot agree more regarding the need for atomistic simulations. For a correct estimation of the energy barriers, we believe such simulations should involve whole nuclei embedded in the matrix – clearly this is something beyond the capabilities of current DFT calculations. This will require extensive calculations using newly developed semi-empirical potentials, which is an endeavour for the future. This point was hinted at in the last paragraph of Sec. 2, Supplementary Information, and has now been further emphasized in the revised manuscript.

Classical nucleation theory indeed provides only a coarse description of the energy landscape for different nucleus configurations. It may be argued whether the continuum concepts of interfacial energy and different strain components can be applied with any accuracy for particles containing only a few hundred atoms. But we believe our CNT calculations provided a useful first approximation for explaining our experimental observations, and in particular for roughly quantifying the different energy contributions for TDN and nucleation from the solid solution. Accurate atomistic simulations of fully embedded nuclei must await significant additional research effort with newly developed semi-empirical potentials.

Nevertheless, we still think that, on balance, the evidence from our experiments and calculations favours a model with fewer vacancies for the θ'' to θ' transformation. The text in the main manuscript and Supplementary Information has been modified to express the above points.

2) The authors demonstrated that vacancy supersaturation promotes heterogeneous nucleation by incorporating vacancy contribution to the nucleation energy through the corresponding thermodynamic driving force (reduction of vacancy supersaturation), an elastic contribution (reduction of the precipitate eigenstrain), and also an interface contribution. There are still some parameters of the model I do not fully understand: could the authors be more precise when defining the corresponding three contributions in Eq. 3 of SI? I cannot see what is exactly the elastic energy variation ΔG_{el} and I do not understand where the interface term comes from as the authors wrote just after Eq. 3 "that the semi-coherent interfacial energy remains the same".

Response:

Further information on the different terms of Eq. 3 has now been included in Sec. 3.2 and a new diagram describing the geometry of TDN has been added as Fig. 16(c).

But, more importantly, this contribution of vacancy supersaturation will also be present for homogeneous nucleation and will also enhance homogeneous nucleation. I don't understand why the authors did not consider this supersaturation also for homogeneous nucleation. At the end, I guess that vacancy supersaturation will generally promote nucleation, either heterogeneous and homogeneous, and that heterogeneous nucleation is favoured over homogeneous nucleation mainly because of elastic and interface effects. This point should be made clearer.

Response:

It is indeed correct that the vacancy contributions considered for TDN should also be considered for other nucleation mechanisms, such as for θ' nucleating directly from the supersaturated solid solution (SSS). This calculation is now described as a new figure (Fig. S27) and a new section (Sec. 3.5) in the Supplementary Information. Not surprisingly, the energy barriers and critical radius for nucleation are reduced somewhat, and 1.5c-thick θ' nuclei are now slightly favoured over 2c-thick θ' nuclei. This is entirely consistent with experimental observations (see Fig. S9). However, this mechanism is likely to remain far less favourable than nucleation of the coherent phase θ'' , which is mainly limited by Cu diffusion, at least for ageing temperature below the θ'' solvus (i.e. $<220^\circ\text{C}$). In the case of nucleation from SSS, the θ' phase is not hampered solely by the lack of vacancies, but by competition from the easily nucleated θ'' phase. In the early stages of ageing, some θ'' phase may transform to θ' via TDN. However, the slow kinetics of the formation of the θ' phase (as evident from hardness curves in early work from Hardy, J. Inst. Metals 80 (1951-52) 483) indicates that θ'' forms very slowly and most θ'' precipitates will be associated with an insufficient number of vacancies for TDN. This is now explained in the revised manuscript.

To summarize, the proposed mechanism for the formation of theta' precipitate looks reasonable but is a possibility among others. The developed model allows to incorporate the different ingredients corresponding to this mechanism to lead to a reasonable thermodynamic description, but one cannot claim that this model demonstrates the validity of this new precipitation mechanism. From my point of view, the question is still open and will require further simulations, probably directly relying on an atomic description.

Response:

As stated above, we agree that more extensive and sophisticated atomistic simulations are required to confirm the proposed mechanisms. However, we believe that the present work constitutes a significant advance in understanding the atomic mechanisms, and more specifically, the role of vacancies, in solid-to-solid phase transformations.

REVIEWERS' COMMENTS:

Reviewer #3 (Remarks to the Author):

All my comments have been addressed.

“Transforming Solid-State Precipitates Via Excess Vacancies” by L. Bourgeois, Y. Zhang, Z. Zhang, Y. Chen and N.V. Medhekar

The Reviewer’s comments and editorial requests are shown in black and the authors’ response to each comment/request is shown in **blue**.

REVIEWERS' COMMENTS:

Reviewer #3 (Remarks to the Author):

All my comments have been addressed.

Thank you.